# Model Tells You Where to Merge: Adaptive KV Cache Merging for LLMs on Long-Context Tasks

## Abstract

Serving LLM inference for long sequences poses significant challenge due to the enormous memory footprint from key-value (KV) cache. In this paper, we propose a novel KV cache merging algorithm, *KVMerger*, to achieve adaptive KV cache compression for long sequence tasks without significant performance degradation. Our approach is inspired by the intriguing observation that key states exhibit high similarity at the token level within a single sequence. Based on the observation, we develop an effective merging set identification algorithm to identify suitable KV states for merging and a Gaussian kernel weighted merging algorithm to selectively merge all states within each merging set. Our merging set identification algorithm stimulates the second observation that KV cache sparsity, from similarity perspective, is independent of the dataset and remains persistent at the model level. We conduct extensive experiments to demonstrate the effectiveness of *KVMerger* for long-context tasks under constrained memory budgets, on Llama2-7B/13B-chat and Mistral-7B-instruct. Our results show that *KVMerger* achieves superior performance across tasks compared to other KV cache compression techniques, including H2O and CaM on LongBench, ZeroScroll, and Needle-in-a-Haystack benchmarks, with both $50\%$ and $35\%$ KV cache budgets.

## 1 Introduction

Large Language Models (LLMs) have demonstrated exceptional performance across a variety of applications, particularly excelling in long-context scenarios that are increasingly relevant in everyday life. However, as LLMs process larger volumes of data over extended contexts, *KV cache* starts to pose a substantial obstacle to LLM's performance and scalability. For example, a 175-billion parameter GPT-3 model, with a batch size of 64 and a sequence length of 4,096 tokens (including both prefilled and generated tokens), necessitates approximately 1,208 GB of GPU memory (Liu et al., 2024), which exceeds the memory capacity of most advanced GPUs. Therefore, compressing KV cache while maintaining LLM accuracy, especially for long-context tasks, becomes quite essential.

Current efforts for KV cache compression can be broadly categorized into three types: quantization, eviction, and merging, as illustrated in Figure 1. Quantization replaces floating point KV states (e.g., FP16) with low-bit values to decrease memory usage. Recent advancements, such as Coupled Quantization (Zhang et al., 2024b) and KIVI (Zirui Liu et al., 2023), have demonstrated that KV cache can be quantized to 1-bit or 2-bit precision while preserving performance. In contrast, KV cache eviction methods selectively remove unimportant states based on certain signals from the model, thereby reducing the memory footprint by limiting the number of key and value states in the KV cache (Xiao et al., 2024; Liu et al., 2023b; Zhang et al., 2023; Ge et al., 2024). While eviction-based methods have demonstrated promising results on short context tasks with simple perplexity metrics, a significant drawback of eviction methods is their potential to accidentally and permanently remove important tokens, leading to context damage and adversely affecting their effectiveness in long-context tasks that heavily rely on context information.

Very recently, KV cache merging has emerged a complementary method of eviction (Zhang et al.). Unlike eviction-based methods, KV cache merging technique does not strictly remove key and value states. Instead, it involves merging states that are otherwise to be dropped by eviction method into single token state. By amalgamating states rather than directly evicting them, this method ensures

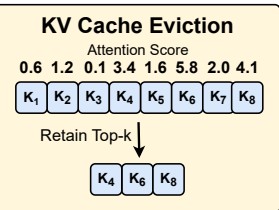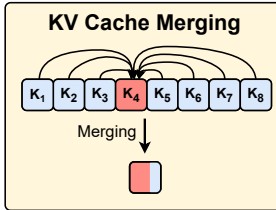

Figure 1: Three categories of KV cache compression techniques: KV cache quantization (left), KV cache eviction (middle), and KV cache merging (right). For the illustration of KV cache eviction, we use aggregated attention scores as the eviction signal, and k is set to 3; for KV cache merging, we illustrate many-to-one merging. The key state in red represents the state which incorporates the information of other remaining states. Value states are processed in the same way as key states.

that essential information not captured by the attention scores is retained. It is noteworthy that, although token merging is well-established in computer vision (CV) (Zeng et al., 2022) (Bolya et al., 2023) (Kim et al., 2023) (Zhang et al., 2024a), the application of key and value states merging in LLMs has not been extensively explored due to several significant challenges. *It is difficult to choose criteria for identifying sets of states to be merged without losing critical information, given that LLM KV caches often have to serve long sequences with high dimensionality.* Additionally, *even after identifying the sets of states to be merged, it remains challenging to find the optimal merging policy, given that there are numerous algorithms for creating merged states.* Effective merging techniques must strike a delicate balance between reducing memory usage and preserving the semantic integrity of the contexts.

To address the aforementioned challenges associated with KV cache merging, we propose an effective KV cache merging method for improving LLMs' performance on long-context tasks. We start by introducing and analyzing an intriguing observation: *key states exhibit high cosine similarity at the token level within a single sequence across different attention heads and model layers.* our observation also opens opportunities for effective merging of key and value states based on their cosine similarity. Subsequently, we formulate the KV cache merging as a constrained clustering problem, and propose an effective merging set identification method for KV cache merging. Based on that, we define KV cache sparsity from the perspective of states similarity. Our finding indicates that KV cache sparsity is independent of the dataset and remains persistent at the model level. Building on top of this, we propose a Gaussian kernel weighted merging function to merge states within each identified merging set. Our contributions can be summarized as follows:

- As one of the pioneering researches concerning KV cache merging for LLMs, we developed *KVMerger*, an effective KV cache merging algorithm especially designed for long-context tasks, including merging set identification and Gaussian kernel weighted merging function.

- We introduce an intriguing observation that key states share a high similarity at the token level within a single sequence, as an important complementary to the previous observations concerning high query states similarity (Dai et al., 2024) and intra-layer KV cache similarity (Liu et al., 2024). We also investigate the root cause of why such phenomenon appears.

- We evaluate *KVMerger* on diverse long-context tasks including LongBench, ZeroScroll, and Needle-in-a-Haystack across various models under different KV cache budgets, and the results show comparable performance or much smaller performance gap compared to full cache budgets.

## 2 RELATED WORK

**KV Cache Quantization.** Quantization methods involve converting high-precision numerical values of key and value states into lower-precision formats, thereby decreasing the storage requirements within the cache (Hooper et al., 2024; Sheng et al., 2023; Liu et al., 2023a; Zhang et al., 2024c). Due to the presence of outliers in key and value states, recent works such as KIVI (Zirui Liu et al., 2023) and Gear (Kang et al., 2024) employ fine-grained group-wise quantization, which quantize small channel groups within each token. MiKV (Yang et al., 2024) addresses the information loss introduced by KV cache eviction methods by preserving those KVs in lower precision rather than directly dropping them. ZipCache (He et al., 2024) proposes an efficient channel-separable quantization scheme, disentangling the channel and token dimensions without excessive memory overhead.

Different from quantization, this work studies compression of KV cache via states merging, which is complementary to quantization.

**KV Cache Eviction.** KV cache eviction methods focus on retaining those important key-value pairs and discard those unimportant ones permanently. One of the common selection policies of key-value pairs is to exploit signals from the attention mechanism of LLMs. For example, H2O (Zhang et al., 2023), Scissorhands (Liu et al., 2023b), and RoCo (Ren & Zhu, 2024) compress KV cache by maintaining a small set of KV states whose corresponding tokens are determined by the ranking of attention scores. StreamingLLM (Xiao et al., 2024) finds that keeping the initial tokens, called attention sink, together with the recent window tokens is pivotal to maintain LLM's performance. More recently, Ge et al. (2024) and Yu et al. (2024) find that attention sinks also occurs in the middle of the sentences, and Ge et al. (2024) introduces FastGen which can choose the most appropriate compression strategy for each heads with different attention distribution patterns.

**KV Cache Merging.** Instead of permanently discarding key and value states, KV cache merging offers a promising direction for KV cache compression while maintaining the performance of LLMs. As the first work concerning cache merging, CaM (Zhang et al.) adaptively merges to-be-evicted value states into the remaining conserved value states, resulting in minimal output perturbation due to the merging operation. Recently, several concurrent work also propose to merge KV cache. For example, MiniCache (Liu et al., 2024) finds that KV states of some consecutive layers have high similarity and proposes an effective intra-layer KV cache merging and restoration algorithms to reduce memory usage by KV cache. Similarly, D2O Wan et al. (2024) selectively merges both value and key states to be evicted with those to be conserved using an Exponential Moving Average (EMA) threshold, and uses weighted merging based on cosine similarity. However, these methods rely heavily on prior eviction strategies, defining effective KV cache merging algorithms remain open challenges. This paper is the first one that formally formulates the KV cache merging problem and conducts in-depth study of this problem via effective merging set identification policy and Gaussian weighted merging algorithm while exploiting the high similarities among key states.

## 3 PROBLEM FORMULATION

Formally, we study the performance impact of LLMs after compressing their KV cache. For a decoder only pre-trained LLM $f$ with $l$ layers, we denote key states and value states for each layer as $\mathcal{K} \in \mathbb{R}^{n \times d}$ and $\mathcal{V} \in \mathbb{R}^{n \times d}$, respectively. Let $Q_t \in \mathbb{R}^{1 \times d}$ denote the query state at time step $t$. Then, the output $\mathcal{O}_t$ for each attention head at a certain layer of $f$ can be formulated as:

$$\mathcal{O}_t = \mathcal{A}_t \mathcal{V}, \ \mathcal{A}_t = softmax\left(\frac{Q_t \mathcal{K}^T}{\sqrt{d_k}}\right) \tag{1}$$

Our primary objective is to develop an efficient many-to-one merging algorithm $M$ for KV cache, which generate merged key states and merged value states, respectively, such that the performance of $f$ with $M$ remains similar or comparable to $f$ with the original uncompressed KV cache.

**Definition 3.1** (KV Cache Merging Problem, informal). *Let $\mathcal{O}_t$ represent the original output of $f$, and let $\mathcal{O}_t^*$ represent the output after merging. $M$ must satisfy the following optimization criterion:*

$$M = \arg\min_I \frac{\sum_l |F_l(I(\mathcal{K}))|}{\sum_l |\mathcal{K}|}, \tag{2}$$

*subject to $|\mathcal{O}_t - \mathcal{O}_t^*| \leq \epsilon$, where $\epsilon$ is an acceptable small positive value, ensuring the degradation in performance is negligible and within acceptable bounds. $M$ also has the following properties:*
- $|M(\mathcal{K})| / |\mathcal{K}| \leq 1, |M(\mathcal{V})| / |\mathcal{V}| \leq 1$
- $|M(\mathcal{K})| / |\mathcal{K}| = |M(\mathcal{V})| / |\mathcal{V}|$ *(make sure key and value states have the same compression ratio)*

The merging algorithm $M$ consists of two parts: (i) a policy $I$ that identifies sub KV cache sets, and (ii) a merging function $F$ that maps the states in each merging set to a single state.

**KV Cache Merging Sets Identification Policy $I$.** We define the identification policy $I$ as:
- $|\mathcal{K}| = |\mathcal{K}_c| + |\mathcal{K}_m|, |\mathcal{V}| = |\mathcal{V}_c| + |\mathcal{V}_m|$
- $|\mathcal{K}_c| = |\mathcal{V}_c|, |\mathcal{K}_m| = |\mathcal{V}_m|$ *(make sure key states and value states come in pair)*

where $\mathcal{K}_c$ and $\mathcal{K}_m$ represent the subsets of key states to be conserved and merged, respectively, and $\mathcal{V}_c$ and $\mathcal{V}_m$ represent the subsets of value states to be conserved and merged, respectively. The above definition is a general formulation. For example, when $|\mathcal{K}_c|$ and $|\mathcal{V}_c|$ are zero, all key and value states are merged, resulting in a full cache without any states eviction.

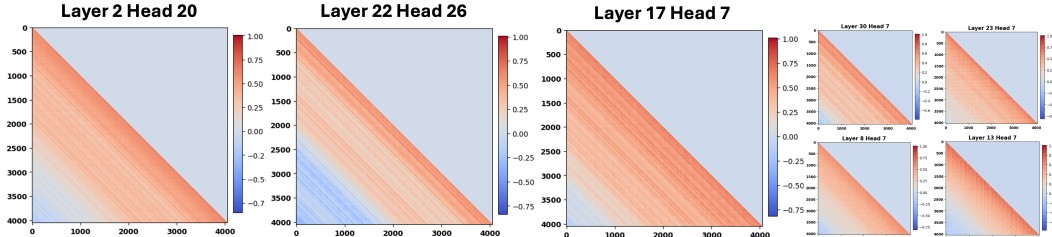

Figure 2: Visualization of the cosine similarity map of key states at the token-wise level produced by running the inference process on the Llama2-7b-chat model by randomly sampling data from the SynthWiki dataset. Observations include: (1) Key states share strong similarity within one sequence across different layers and heads; (2) The similarity between key states has the property of locality, i.e., adjacent tokens exhibit higher similarity. More visualizations are shown in appendix A.2.

**KV Cache Merging Function $F$.** We define the merging function $F$ such that
$$F : \{S_i\}_{i=1}^{|\mathcal{K}_m|} \to \{s_i^*\}_{i=1}^{|\mathcal{K}_m|} \quad \text{, where} \quad F(S_i) = s_i^*, \quad i = 1, 2, \ldots, |\mathcal{K}_m|$$
where $S_i$ represents each merging set, and $s_i^*$ is the merged new state for each merging set. $|\mathcal{K}_m|$ is the total number of merging sets as previously defined.

## 4 OBSERVATIONS

In this section, we present two key observations illustrating that KV cache sparsity is universal for long-context tasks when viewed from the perspective of state similarity. These observations form the basis for our development of the adaptive KV cache merging algorithm *KVMerger*.

### 4.1 KV CACHE SIMILARITY

Inspired by Dai et al. (2024), which reveals the phenomenon that query states share significant similarity at the token level in LLMs, we observe for the first time that *key states also exhibit very high similarity at the token level within single sequence*. We will first demonstrate the generalization of this token level similarity in key states and then analyze the potential reasons behind it.

**Observation: key states exhibit high, localized token-level similarity.** We conduct the inference process on the Llama2-7b-chat model by randomly sampling data from the SynthWiki dataset (Peysakhovich & Lerer, 2023) with average sequence length being about 4K. Then, we visualize the cosine similarity of key states at the token level within a sequence using the following equation:
$$similarity\,(k_i, k_j) = \frac{k_i k_j^T}{||k_i|| \cdot ||k_j||}, \quad 1 \le i, j \le T, \tag{3}$$
where $T$ is the total length of the sequence. $k_i$ represents the $i$-th key state, and $k_j$ represents the $j$-th key state. The results are shown in Figure 2. We observe that the similarity maps illustrate a clear oblique color segmentation. The closer it is to the diagonal, the more intense the color becomes, indicating that key states exhibit a strong localized similarity as query states do (Dai et al., 2024). Moreover, we observe from Figure 3(a) that the local similarity between one key state and the other consecutive key states shows different fluctuations for different attention heads. We also examine the cosine similarity of value states but do not observe the same property. One interesting question arises: *why do such localized token similarity exhibit in key states, while value states do not?*

**Analysis.** A key difference between key states, query states, and value states is that Rotary Position Embedding (RoPE) (Su et al., 2023) is applied to the key and query states, but not to the value states. We hypothesize that this application of RoPE is the primary factor contributing to the observed differences in token-level similarity. Specifically, by rotating the embeddings in a multi-dimensional space, RoPE effectively captures the relative positions and order of tokens within a sequence. If we denote two adjacent input tokens as $x_m, \; x_n \in \mathbb{R}^d$ where $m$ and $n$ are two random integers, and $0 \le j \le (d-1)/2$. Then the position information of each token is incorporated by RoPE via the following equations:
$$k_{m,[2j:2j+1]} \;=\; W_k x_m e^{im\theta_j}, \; k_{n,[2j:2j+1]} = W_k x_n e^{in\theta_j}, \; \theta_j = b^{\frac{-2j}{d}}, \tag{4}$$
where $W_k$ is the matrix for key projection, and $e^{in\theta_j}$ is the rotary embedding component. $b$ is called as the rotary base, which is set to 10000 by default (Su et al., 2023). Then, we introduce two lemmas

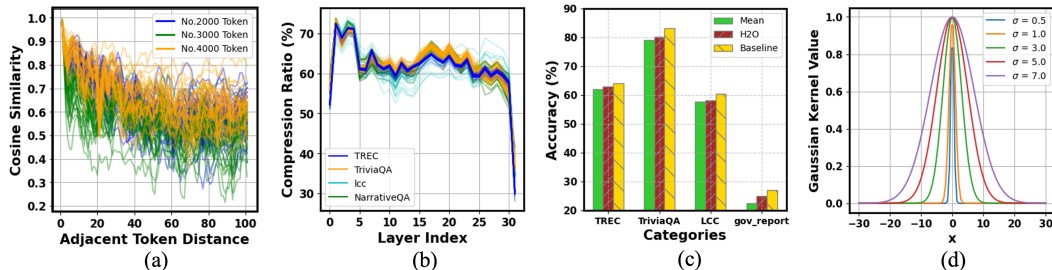

Figure 3: (a): The cosine similarity changes between the current token and its adjacent tokens across distinct attention heads and layers. We show the above changes for tokens with indices being 2000, 3000, and 4000.(b) The layer-wise compression ratios obtained by our proposed merging set identification algorithm for different samples and different tasks. (c) The comparison of long-context performance between H2O and average weighted merging with our proposed merging set identification algorithm. (d) The illustration of Gaussian kernel function with different values of $\sigma$.

that clearly show the conditions under which two token embeddings exhibit high cosine similarity after the application of the RoPE operation.

**Lemma 4.1** (Informal). *Consider two vectors $\mathbf{k}_m$, $\mathbf{k}_n \in \mathbb{R}^{1 \times d}$. If their cosine similarity is 1, then the cosine similarity of any $1 \times 2$ vectors, $\mathbf{k}_{m,j} = [k_{m,2j}, k_{m,2j+1}]^T$ and $\mathbf{k}_{n,j} = [k_{n,2j}, k_{n,2j+1}]^T$, formed by the $2j$-th and $(2j+1)$-th elements of $\mathbf{k}_m$ and $\mathbf{k}_n$, $0 \le j \le (d-1)/2$, is also equal to 1.*

**Lemma 4.2** (Informal). *Consider integer $j$ such that $0 \le j \le \frac{d-1}{2}$. Define the vectors $\mathbf{k}_{m,j}$ and $\mathbf{k}_{n,j}$ as $\mathbf{k}_{m,j} = [k_{m,2j}, k_{m,2j+1}]^T$ and $\mathbf{k}_{n,j} = [k_{n,2j}, k_{n,2j+1}]^T$, and define the vectors $\mathbf{k}'_{m,j}$ and $\mathbf{k}'_{n,j}$ as $\mathbf{k}'_{m,j} = \mathbf{k}_{m,j}/e^{im\theta_j}$ and $\mathbf{k}'_{n,j} = \mathbf{k}_{n,j}/e^{in\theta_j}$. If $similarity\,(\mathbf{k}_{m,j}, \mathbf{k}_{n,j}) = 1$, we have:*

$$\cos{(m-n)} < \frac{\langle \mathbf{k}'_{m,j}, \mathbf{k}'_{n,j}\rangle}{\|\mathbf{k}'_{m,j}\| \cdot \|\mathbf{k}'_{n,j}\|} \le 1,$$

*where $\langle \mathbf{k}'_{m,j}, \mathbf{k}'_{n,j}\rangle$ denotes the inner product of $\mathbf{k}'_{m,j}$ and $\mathbf{k}'_{n,j}$, and $\|\mathbf{k}'_{m,j}\|$ and $\|\mathbf{k}'_{n,j}\|$ denote the norms of $\mathbf{k}'_{m,j}$ and $\mathbf{k}'_{n,j}$, respectively.*

The complete proof of the above lemma is shown in appendix A.1. The conclusions of lemmas 3.1 and 3.2 are the necessary conditions of $similarity(\mathbf{k}_{m,j}, \mathbf{k}_{n,j}) = 1$. A cosine similarity of $\mathbf{k}'_{m,j}$ and $\mathbf{k}'_{n,j}$ falling beyond the range $(\cos{(m-n)}, 1]$ will result in the failure of $similarity(\mathbf{k}_{m,j}, \mathbf{k}_{n,j}) = 1$. The analysis above clarifies why value states exhibit low similarity at the token level. Without the RoPE operation, value states are incapable of achieving rotation to comparable angles. The above analysis also indicates that merging highly similar key states is sensible.

### 4.2 PERSISTENT KV CACHE SPARSITY

We have shown that key states within a sequence exhibit significant token-level similarity in LLMs. Leveraging this, we apply a specialized greedy clustering algorithm to group consecutive key states with similarity values above a certain threshold, processing from the last to the first token. This yields a set of groups, each containing highly similar consecutive key states, forming a merging set. The number of this merging set corresponds to the number of key states after merging. Details of this set identification algorithm will be formally introduced in Section 5.1.

**Observation: The KV cache sparsity for different samples are persistent at the model level.** Figure 3(a) shows that the similarity distributions of different tokens vary across distinct attention heads and layers. The size of each subset of key states is governed by the similarity threshold defined. Lowering the threshold results in the inclusion of a larger number of key states within a single merging set, thereby leading to varied compression ratios across all attention heads and layers. To investigate the actual compression ratio achieved by the previous set identification algorithm, we conduct inference processes on the Llama2-7b-chat model. This involves randomly sampling 200 instances from the subset of LongBench (Bai et al., 2024) tasks and calculating the average compression ratio for each layer, as shown in Figure 3(b). We observe that the layer-wise compression ratios were highly consistent across different samples from the same task and even across different tasks. This intriguing finding suggests that the *kv cache sparsity, resulting from the high similarity*

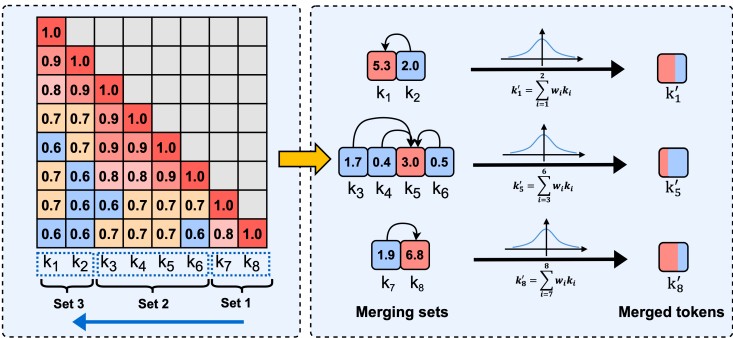

Figure 4: The whole framework of *KVMerger* is comprised of two major modules. The first module is to identify the merging set. The toy similarity map is used to illustrate this process, and the threshold for cosine similarity is set to 0.8. The second module is to merge key and value states within each identified merging set via Gaussian kernel weighted merging. The key state in red color represents the pivotal key state, where all the remaining key states should be weighted merged to that one. The values on key states represent the aggregated attention scores.

*exhibited by key states, is independent of the dataset and remains persistent at the model level.* The same static KV cache sparsity property can be observed in other models as Appendix A.3 shown.

**Insights** The observed static KV cache sparsity suggests that it is possible to determine the layer-wise compression ratios by adjusting the cosine similarity threshold, making it convenient for us to control the total compression ratio for a certain model, thereby reducing the KV cache memory.

## 5 PROPOSED ADAPTIVE KV MERGING ALGORITHM

In this section, we propose *KVMerger*, an adaptive KV merging algorithm, for LLMs based on the above observations. The whole pipeline of *KVMerger* is depicted in Figure 4. We first introduce the merging set identification algorithm in Section 5.1, which can be viewed as solving a constrained clustering problem. We propose a specialized greedy clustering algorithm to solve this. In Section 5.2, we delineate Gaussian kernel weighted merging, which is a many-to-one states merging algorithm without introducing significant information loss.

### 5.1 GREEDY POLICY FOR MERGING SET IDENTIFICATION

To solve the merging set identification problem described in Section 3, we regard it as a variant of clustering problem, which we define below:

**Definition 5.1** (Constrained Clustering Problem for KV Cache Merging, formal). *Given the key states to be merged as $\mathcal{K}_m = \{k_1, k_2, \ldots, k_n\}$ from a certain attention head at a certain layer of $f$, where each $k_n \in \mathbb{R}^{1 \times d}$, and a similarity function $\delta : \mathcal{K}_m \times \mathcal{K}_m \to \mathbb{R}^{n \times n}$, and a cosine similarity threshold $\theta$, partition $\mathcal{K}_m$ into $c$ merging sets $\{\mathcal{S}_1, \mathcal{S}_2, \ldots, \mathcal{S}_c\}$ such that the intra-cluster similarity is maximized and the inter-cluster similarity is minimized, which can be expressed as:*

$$\max_{\mathcal{S}_1, \mathcal{S}_2, \ldots, \mathcal{S}_c} \left( \sum_{i=1}^{c} \sum_{k_a, k_b \in \mathcal{S}_i} \delta(k_a, k_b) - \sum_{(i,j), i \neq j} \sum_{k_a \in \mathcal{S}_i, k_b \in \mathcal{S}_j} \delta(k_a, k_b) \right)$$

- *Each merging set $\mathcal{S}_i$ should satisfy: $\mathcal{S}_i \cap S_j = \emptyset$   for   $i \neq j$,   and   $\bigcup_{i=1}^{c} \mathcal{S}_i = \mathcal{K}_m$;*
- *$\forall \mathcal{S}_i, \exists j$ such that $\mathcal{S}_i = \{k_j, k_{j+1}, \cdots, k_{j+|\mathcal{S}_i|-1}\}$.*

The similarity function, $\delta$, we used here is cosine similarity based on the observation in Section 4.1. In order to conserve the locality similarity property of key states, the merging set identification problem is a constrained clustering problem, meaning that all elements in one cluster are expected be consecutive, and we do not merge states with high similarity but far away from each other for simplicity. Then, we propose a specialized greedy clustering algorithm to find all merging sets shown as Algorithm 1, which follows the above requirements.

*KVMerger* also retains the KV states whose corresponding aggregated attention scores fall within the top-k range, including both attention sinks and heavy-hitters, which represent the most impor-

tant and frequently accessed elements by LLMs. We assume that those key and value states are quite sensitive to merging and cannot participant in merging process to avoid information damage.

| **Algorithm 1** Merging Set Identification | **Algorithm 2** Merging Policy |
|---|---|
| 1: **procedure** CLUSTER($\mathcal{K}_m, \delta, \theta$) | 1: **procedure** MERGE($\mathcal{S}_c, A$) |
| 2:     Initialize empty lists $G$ and $C$, $j = \|\mathcal{K}_m\|$ | 2:     Compute aggregated attention score for |
| 3:     Add $k_j$ to $C$ |       each state in the merging set $\mathcal{S}_c$: |
| 4:     **for** $i = j - 1$ **to** 1 **do** |       $\mathbf{a}_i = \sum_{i=1}^{\|\mathcal{S}_c\|} A\,[i,:]$ |
| 5:       **if** $\delta(k_i, k_j) > \theta$ **then** | 3:     Find pivotal state: $p = \underset{i \in \mathcal{S}_c}{\arg\max}\,(\mathbf{a}_i)$ |
| 6:         Add $k_i$ to $C$ | |
| 7:       **else** | 4:     **for** $j = 1$ **to** $\|\mathcal{S}_c\|$ **do** |
| 8:         Add $C$ to $G$, $j = i$ | 5:       $\mathbf{g}_{pj} = \mathcal{G}(\mathbf{k}_p, \mathbf{k}_j)$ |
| 9:         Initialize a new $C$ with $k_j$ | 6:     **end for** |
| 10:      **end if** | 7:     $\mathbf{w}_j = \mathbf{g}_{pj}/\sum \mathbf{g}_{pj}$ |
| 11:     **end for** |       $k_M = \sum \mathbf{w}_j k_j$ |
| 12:     **return** $G$ | 8:     **return** $k_M$ |
| 13: **end procedure** | 9: **end procedure** |

## 5.2 Gaussian Kernel Weighted Merging

**Definition 5.2** (Weighted KV cache Merging, formal). *Given identified merging sets of key states and value states as $\mathcal{S}_k = \{k_i, k_{i+1}, \ldots, k_p, \ldots k_{i+n}\}$ and $\mathcal{S}_v = \{v_i, v_{i+1}, \ldots, v_p, \ldots v_{i+n}\}$, where $k_p$ and $v_p$ denote the pivotal key state and pivotal value state, respectively. Then, the weighted merging key states and value states can be defined as:*

$$k_p^* = w_p k_p + \sum_{k_i \in \mathcal{S}_k, i \neq p} w_i k_i, \quad v_p^* = w_p v_p + \sum_{v_i \in \mathcal{S}_v, i \neq p} w_i v_i \qquad (5)$$

*where $w_p$ and $w_i$ denote the weight assigned to the pivotal state and other states in the merging set.*

We define the weighted merging function for KV cache merging in Definition 5.2, which follows the many-to-one merging definitions from Wan et al. (2024). In terms of Definition 5.2, two principal design factors directly influence merging efficacy. The first factor is the selection of the pivotal state, to which all other states are merged. The second factor involves the assignment of weights to each state, with the pivot state having the largest weight to preserve the information.

**Selection for Pivotal State** We follow previous token eviction methods that using aggregated attention score to select pivotal token as it indicates the importance of tokens, which can be expressed as:

$$\mathbf{k}_p = \underset{i \in S_c}{\arg\max}\,(\mathbf{a}_i), \quad \mathbf{a}_i = \sum_{i=0}^{\|S_c\|} A\,[i,:], \qquad (6)$$

where $A$ is the attention map of the whole sequence. Note that the index of pivotal token for value states within each merging set is the same as key states.

**Gaussian Kernel Weights** Initially, we use average weighted method to merge all states for each merging set. However, LLM's performance is sub-optimal via this simple method as shown in Figure 3(c). To reduce the influence of dissimilar or distant states, we utilize Gaussian kernel weighted merging, which is expressed as:

$$\mathbf{g}_{pi} = \mathcal{G}(\mathbf{k}_p, \mathbf{k}_i) = exp\left(-\frac{\|\mathbf{k}_p - \mathbf{k}_i\|^2}{2\sigma^2}\right), \quad \sigma = \frac{\sum_{i=0}^{\|\mathbf{S}_c\|} \|\mathbf{k}_p - \mathbf{k}_i\|^2}{\sqrt{2}\|\mathbf{S}_c\|}. \qquad (7)$$

Gaussian kernel is able to assign greater weight to elements that are nearer to the pivotal state, offering a smooth and flexible weighting that reduces noise and outlier impact. This local weighting characteristic ensures that the merged result is significantly shaped by nearby states, maintaining local structure. Then, the merging weights for key states and value states can be formalized as:

$$\mathbf{w}_i = \frac{\mathbf{g}_{pi}}{\sum_{j=0}^{\|S_c\|} \mathbf{g}_{pj}}, \quad \mathbf{w}_p = \frac{1}{\sum_{j=0}^{\|S_c\|} \mathbf{g}_{pj}}. \qquad (8)$$

As demonstrated in Definition 5.2, the weight assigned to each $\mathbf{k}_i$ and $\mathbf{v}_i$ is directly governed by the squared $l_2$ norm between the pivotal token and the remaining tokens. This indicates that if $\mathbf{k}_i$ is close to $\mathbf{k}_p$ in the Euclidean space, more weight will be assigned to $\mathbf{k}_i$ as Figure 3(d) illustrates.

Table 1: *KVMerger* for Llama2-7B/13B-chat and Mistral-7B-Instruct on **LongBench** datasets.

| models | budget | method | 2wikimqa | gov_report | narrativeqa | pr_en | multifieldqa_en | trec | multi_news | triviaqa | qasper | avg. |
|---|---|---|---|---|---|---|---|---|---|---|---|---|
| Llama2-7B-chat | 100% | Full Cache | 31.45 | 26.99 | 18.74 | 8.00 | 36.60 | 64.00 | 26.26 | 83.09 | 21.83 | 35.22 |
| | 50% | H2O | 29.96 | 24.86 | 17.48 | 7.00 | 33.58 | 63.50 | 26.00 | 82.51 | **21.04** | 34.00 |
| | | CaM | 30.69 | 24.46 | 17.08 | 6.50 | 33.98 | 63.50 | 24.66 | 82.17 | 20.00 | 33.67 |
| | | *KVMerger* | **32.99** | **25.31** | **18.50** | 7.33 | **36.89** | **64.00** | **26.29** | **83.62** | 20.04 | **35.02** |
| | 35% | H2O | 30.57 | 24.48 | 17.85 | 7.00 | 32.17 | 63.00 | 25.37 | 80.89 | 20.04 | 33.49 |
| | | CaM | 31.06 | 23.80 | 18.36 | 6.00 | 33.07 | 62.50 | 25.23 | 81.86 | 18.37 | 33.36 |
| | | *KVMerger* | **32.29** | **25.24** | **19.12** | 7.00 | **33.82** | **63.50** | **25.64** | **82.76** | **21.09** | **34.50** |
| Llama2-13B-chat | 100% | Full Cache | 13.21 | 27.59 | 14.42 | 15.25 | 27.44 | 68.50 | 26.69 | 87.42 | 17.15 | 33.07 |
| | 50% | H2O | 13.39 | 26.20 | **15.01** | 15.50 | 26.40 | 68.00 | 25.35 | 84.73 | 17.10 | 32.40 |
| | | CaM | 13.30 | 25.88 | 13.47 | 15.00 | 26.96 | 67.50 | 26.06 | 84.65 | 16.58 | 32.16 |
| | | *KVMerger* | **13.46** | **26.63** | 14.4 | **16.00** | **27.29** | **68.50** | **26.12** | **87.48** | **17.22** | **33.01** |
| | 35% | H2O | 12.26 | 25.52 | 13.14 | **14.50** | 25.75 | 67.50 | 25.59 | 83.53 | **16.35** | 31.57 |
| | | CaM | **13.43** | 25.37 | 13.58 | 12.50 | 25.70 | 67.50 | 25.04 | 84.95 | 16.34 | 31.60 |
| | | *KVMerger* | 12.61 | **26.12** | **13.60** | 14.00 | **26.75** | **68.00** | **26.32** | **86.76** | 16.24 | **32.27** |
| Mistral-7B-Instruct | 100% | Full Cache | 31.47 | 26.55 | 21.96 | 25.00 | 39.50 | 61.00 | 26.44 | 83.89 | 30.12 | 38.44 |
| | 50% | H2O | 29.21 | 19.91 | 17.65 | 8.00 | 25.50 | 53.00 | 19.95 | 74.55 | 21.51 | 29.92 |
| | | CaM | 29.57 | 22.67 | 19.43 | 12.00 | 28.95 | 58.00 | 20.17 | 81.82 | 21.87 | 32.72 |
| | | *KVMerger* | **32.44** | **24.05** | **21.85** | **23.00** | **31.23** | **60.00** | **20.87** | **84.16** | **24.52** | **35.79** |
| | 35% | H2O | 12.30 | 5.16 | 3.64 | 0.62 | 11.95 | 37.50 | 18.99 | 17.08 | 14.05 | 13.48 |
| | | CaM | 28.77 | 18.70 | 17.76 | 8.50 | 25.31 | 45.50 | 19.72 | 72.88 | 17.25 | 28.27 |
| | | *KVMerger* | **30.77** | **20.99** | **23.58** | **23.50** | **28.10** | **60.5** | **19.94** | **83.82** | **24.13** | **35.04** |

Specifically, if $\sigma$ approaches 0 and $||\mathbf{k}_p - \mathbf{k}_i||^2$ is significantly different from 0, the weight assigned to $\mathbf{k}_i$ tends towards 0. We empirically define $\sigma$ as the mean value of $\mathbf{g}_{pi}$ for all tokens within each merging set to avoid such situation. The complete merging policy is described by Algorithm 2.

# 6 EXPERIMENT

## 6.1 EXPERIMENTAL SETTINGS

**Models and Tasks** We evaluate *KVMerger* using three models: Llama2-7B/13B-chat (Touvron et al., 2023) and Mistral-7B-Instruct-v1.0(Jiang et al., 2023). Our evaluation focuses primarily on instruction-tuned models, as these are meticulously optimized for dialogue use cases and question-answering scenarios. The above three models are evaluated on two commonly used benchamrks for long-context scenario, that is, LongBench (Bai et al., 2024) and ZeroScrolls (Shaham et al., 2023). Specifically, we use nine datasets in LongBench: 2WikiMQA, gov_report, NarrativeQA, passage_retrieval_en, MultifieldQA_en, TREC, multi_news, TriviaQA, qasper. We use seven datasets in ZeroScrolls: gov_report, SummScreenFD, QMSum, SQuALITY, Qasper, NarrativeQA, Book-SumSort. Additionally, we also individually test our methods on RAG tasks with the Needle-in-a-Haystack test (Guerreiro et al., 2023). The performance of our method for LLMs on all the above tasks are also compared with existing eviction method H2O and merging method CaM.

**Implementation details** We test *KVMerger* in two compression scenarios. The first one is $50\%$ KV cache budget, where the proportion of recent tokens to be reserved is set to $0.17\%$, and the proportion of states not selected for the merging process in terms of aggregated attention scores is set to $0.12\%$. The remaining key states and value states participate in the merging process. The second compression scenario is $35\%$ KV cache budget, where the proportion of recent tokens is set to $0.08\%$, and the proportion of states not selected for the merging process is set to $0.02\%$. The cosine similarity threshold for both two scenarios is set to $0.75$. We conducted our experiments on a cluster with A100 40GB GPUs and a cluster with A100 80GB GPUs. The evaluation process for LongBench and ZeroScrolls follows THUDM (2024) and Lab (2024). The implementation of Needle-in-a-Haystack test follows Kamradt.

## 6.2 EXPERIMENTAL RESULTS ON LONG-CONTEXT TASKS

**LongBench Results** The evaluation results of nine selected LongBench datasets on Llama2-7B/13B-chat and Mistral-7B-Instruct-v1.0 are shown in Table 1. We compare the current KV cache compression methods, including H2O, CaM, with our proposed KV merging method *KVMerger* by preserving both $50\%$ and $35\%$ of contexts in the KV cache. Our results demonstrate that *KVMerger*

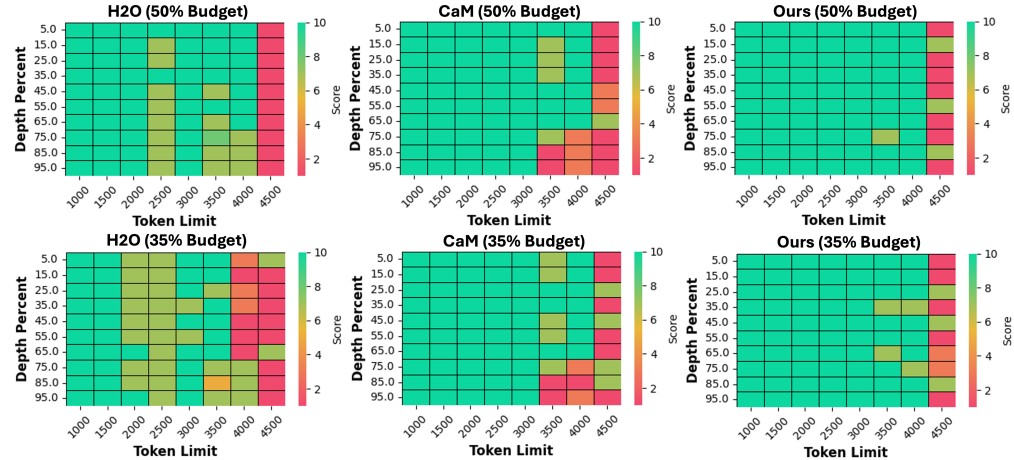

Figure 5: The visualization of needle-in-a-haystack test on Llama2-7B-chat with different KV cache compression methods. The x-axis represents the length of contexts, and the y-axis represents the document depth where the needle is inserted.

consistently outperforms the other KV cache compression techniques across nearly all selected datasets from LongBench. Notably, the performance gaps between our algorithm and the full KV cache scenario for both Llama2-7B/13B-chat and Mistral-7B-Instruct-v1.0 are significantly smaller than the other KV compression methods. Another interesting finding is that the latest value states merging method, CaM, does not perform well on long-context tasks. This may be attributed to the information loss results from eviction of key states, despite the merging of value states.

Note that Mistral-7B-Instruct model leverages Grouped-Query-Attention (GQA) to optimize KV cache memory usage, where each key state corresponds to four query states. When applying H2O to each key state, rather than duplicating states, we use a single attention map. This attention map is generated by averaging the values of four attention maps formed by the four query states, which determines the states to be evicted. For *KVMerger*, we also utilize the same attention map to select pivotal states, ensuring a fair comparison. Our results indicate a significant performance drop for Mistral-7B-Instruct-v1.0 when using H2O. Conversely, *KVMerger* demonstrates the smallest performance decline under both 35% and 50% KV cache budgets, highlighting its efficiency on GQA.

**ZeroScrolls Results** We also evaluate Llama2-7B-chat on ZeroScrolls datasets using different KV cache compression techniques, which are shown as Table 6 in Appendix A.4. Table 6 demonstrates that our proposed KV cache merging method effectively restores the performance of the Llama2-7B-chat model across all selected ZeroScrolls datasets under both 35% and 50% cache budgets. This suggests that *KVMerger* not only mitigates performance degradation but also optimizes the model's handling of extensive data sequences that approach the model's maximum context window.

**Needle In A Haystack Results** We also conduct a detailed comparison of *KVMerger* with other KV cache compression techniques on retrieval tasks using the needle-in-a-haystack test. This test involves placing a random fact in the middle of a long context window and assessing the model's ability to retrieve this statement across varying document depths and context lengths. Specifically, we test Llama2-7B-chat on document depths ranging from 5% to 95% and context lengths ranging from 1000 to 4500 tokens under both 35% and 50% cache budgets. The corresponding results are illustrated as Figure 5. Our findings indicate that both CaM and our merging algorithm outperform the eviction method H2O. However, our proposed method achieves the highest retrieval performance, consistently delivering high scores across various context lengths and depth percentages.

**Memory Usage Analysis** We analyze the inference memory consumption of KVMerger under full cache, 35%, and 50% KV cache budgets. Specifically, we use the Llama2-7B/13B-chat models with an input length of 3,900 and generate 1,024 tokens on a single A100 80GB GPU. During inference, we measure both the peak memory and KV cache memory usage, as presented in Table 2. Our results show that *KVMerger* intuitively reduces KV cache memory consumption by up to 60% under 35% KV cache budget. Additionally, it also decreases peak memory usage during the generation process. These findings demonstrate that *KVMerger* significantly reduces KV cache memory without incurring any additional memory overhead.

Table 2: KV cache and peak memory usage for Llama2 Models

| Model | Scenario | KV Cache | Peak Memory |
|---|---|---|---|
| **Llama2-7B-chat** | Full Cache | 2.00 GB | 18.83 GB |
| | KVMerger 50% | 1.02 GB | 17.90 GB |
| | KVMerger 35% | 0.65 GB | 17.55 GB |
| **Llama2-13B-chat** | Full Cache | 3.13 GB | 32.21 GB |
| | KVMerger 50% | 1.56 GB | 31.67 GB |
| | KVMerger 35% | 1.22 GB | 31.30 GB |

## 6.3 ABLATION STUDY

**Choice of Pivotal State in Gaussian Kernel Weighted Merging** As mentioned in Section 5.2, the selection of pivotal state for each merging set is directly related to the performance of *KVMerger*. To show the significance of defining the pivotal state as our method, we compare it with randomly selecting pivotal state within each merging set by using Llama2-7B-chat model with $50\%$ cache budget. The comparison is shown in Table 3, from which we can see that randomly selecting pivotal states are detrimental to LLMs' performance on long-context tasks.

Table 3: *KVMerger* with different methods of pivotal states selection.

| Pivotal State | 2wikimqa | gov_report | narrativeqa | pr_en | multifieldqa_en | avg. |
|---|---|---|---|---|---|---|
| **Ours** | 32.99 | 25.31 | 18.50 | 7.33 | 36.89 | 24.20 |
| **Random** | 30.01 | 24.07 | 17.72 | 6.50 | 33.30 | 22.12 |

**Choice of Merging Policy** We also compare our proposed Gaussian weighted merging algorithm with average weighted merging to illustrate the significance of Gaussian weighted merging. Specifically, we evaluate LongBench tasks on Llama2-7B-chat model with average weighted merging with $50\%$ cache budget, and the results shown in Table 4. We can see that assigning all states with the same merging weights dramatically degrade the performance. One reasonable explanation is that average merging introduces information distortion by treating every states equally.

Table 4: *KVMerger* with different merging policy.

| $\sigma$ | 2wikimqa | gov_report | narrativeqa | pr_en | multifieldqa_en | avg. |
|---|---|---|---|---|---|---|
| **Gaussian** | 32.99 | 25.31 | 18.50 | 7.33 | 36.89 | 24.20 |
| **average** | 27.04 | 24.96 | 17.50 | 6.34 | 33.37 | 21.84 |

**Choice of $\sigma$ in Gaussian Kernel Weights** In section 5.2, we set $\sigma$ as the mean of $l_2$ distance between the pivotal state and all the remaining states within each merging set. To verify the effectiveness of the proposed $\sigma$ computation method, we apply KVMerger on Llama2-7B-chat model with various $\sigma$ values, and evaluation results are shown as Table 5. We can find that when $\sigma$ is computed by using our method, *KVMerger* demonstrate the optimal results in terms of the overall performance compared with other $\sigma$ values.

Table 5: *KVMerger* with different $\sigma$ values under $50\%$ cache budget.

| $\sigma$ | 2wikimqa | gov_report | narrativeqa | pr_en | multifieldqa_en | avg. |
|---|---|---|---|---|---|---|
| **1** | 31.48 | **25.52** | **18.98** | 6.25 | 36.59 | 23.76 |
| **3** | 30.84 | 25.19 | 18.51 | 4.67 | 37.48 | 23.34 |
| **ours** | **32.99** | 25.31 | 18.50 | **7.33** | **36.89** | **24.20** |
| **6** | 31.69 | 25.39 | 18.45 | 7.83 | 35.82 | 23.84 |

## 7 CONCLUSION

In this paper, we propose *KVMerger*, an adaptive KV cache merging method inspired by the observation that key states exhibit high and persistent similarity within each sequence, allowing for layer-wise KV cache compression. We initially abstract the merging set identification problem as a constrained clustering problem and introduce a specialized greedy clustering algorithm to identify merging sets based on cosine similarities between key states. Furthermore, we implement a Gaussian Kernel weighted merging method to merge key and value states within each merging set. Compared to other KV cache eviction and merging methods, our approach achieves superior results on the LongBench and ZeroScrolls benchmark under the same cache budget. Additionally, our method effectively recovers the model's long-context retrieval capabilities, as demonstrated by the needle-in-a-haystack tests.

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

## A  APPENDIX

### A.1  THEORETICAL ANALYSIS

We present the formal and complete proofs of Lemma 4.1 and Lemma 4.2, which establish the cosine similarity condition for any two key state embeddings prior to the RoPE operation, given that their cosine similarity is equal to 1.

**Lemma 4.1** (Formal version of Lemma 4.1). *Consider two vectors* $\mathbf{k}_m$, $\mathbf{k}_n \in \mathbb{R}^{1 \times d}$. *If their cosine similarity is* 1*, then the cosine similarity of any* $1 \times 2$ *vectors,* $\mathbf{k}_{m,j} = [k_{m,j}, k_{m,2j+1}]^T$ *and* $\mathbf{k}_{n,j} = [k_{n,2j}, k_{n,2j+1}]^T$*, formed by the* $2j$*-th and* $(2j+1)$*-th elements of* $\mathbf{k}_m$ *and* $\mathbf{k}_n$, $0 \le j \le \frac{d-1}{2}$, *is also equal to* 1.

*Proof.* Since

$$similarity\left(\mathbf{k}_m, \mathbf{k}_n\right) = 1, \tag{9}$$

$\mathbf{k}_m$ and $\mathbf{k}_n$ are collinear. Therefore,

$$\mathbf{k}_m = \alpha \mathbf{k}_n, \tag{10}$$

where $\alpha$ is a scalar. It means

$$k_{m,2j} = \alpha k_{n,2j}, \tag{11}$$
$$k_{m,2j+1} = \alpha k_{n,2j+1}. \tag{12}$$

So,

$$[k_{m,2j}, k_{m,2j+1}]^T = \alpha [k_{n,2j}, k_{n,2j+1}]^T. \tag{13}$$

As a result,

$$similarity\left(\mathbf{k}_{m,j}, \mathbf{k}_{n,j}\right) = 1 \tag{14}$$

**Lemma 4.2** (Formal version of Lemma 4.2). *Consider integer* $j$ *such that* $0 \le j \le \frac{d-1}{2}$. *Define the vectors* $\mathbf{k}_{m,j}$ *and* $\mathbf{k}_{n,j}$ *as* $\mathbf{k}_{m,j} = [k_{m,2j}, k_{m,2j+1}]^T$ *and* $\mathbf{k}_{n,j} = [k_{n,2j}, k_{n,2j+1}]^T$, *and define the vectors* $\mathbf{k}'_{m,j}$ *and* $\mathbf{k}'_{n,j}$ *as* $\mathbf{k}'_{m,j} = \mathbf{k}_{m,j}/e^{im\theta_j}$ *and* $\mathbf{k}'_{n,j} = \mathbf{k}_{n,j}/e^{in\theta_j}$. *If* $similarity\left(\mathbf{k}_{m,j}, \mathbf{k}_{n,j}\right) = 1$, *we have:*

$$\cos\left(m-n\right) < \frac{\langle \mathbf{k}'_{m,j}, \mathbf{k}'_{n,j} \rangle}{\|\mathbf{k}'_{m,j}\| \cdot \|\mathbf{k}'_{n,j}\|} \le 1,$$

*where* $\langle \mathbf{k}'_{m,j}, \mathbf{k}'_{n,j} \rangle$ *denotes the inner product of* $\mathbf{k}'_{m,j}$ *and* $\mathbf{k}'_{n,j}$, *and* $\|\mathbf{k}'_{m,j}\|$ *and* $\|\mathbf{k}'_{n,j}\|$ *denote the norms of* $\mathbf{k}'_{m,j}$ *and* $\mathbf{k}'_{n,j}$, *respectively.*

*Proof.* Since $j$ is an integer obeying $0 \le j \le \frac{d-1}{2}$, so

$$-1 < \frac{-2j}{d} \le 0. \tag{15}$$

And $b$ is set to be 10000 by default Su et al. (2023). Therefore,

$$0 < b^{\frac{-2j}{d}} \le 1, \tag{16}$$

which means

$$0 < \theta_j \le 1. \tag{17}$$

Now, focus on the similarity between $\mathbf{k}_{m,j}$ and $\mathbf{k}_{n,j}$, and

$$\frac{\langle \mathbf{k}_{m,j}, \mathbf{k}_{n,j} \rangle}{\|\mathbf{k}_{m,j}\| \cdot \|\mathbf{k}_{n,j}\|} = \frac{\langle \mathbf{k}'_{m,j} e^{im\theta_j}, \mathbf{k}'_{n,j} e^{in\theta_j} \rangle}{\|\mathbf{k}'_{m,j} e^{im\theta_j}\| \cdot \|\mathbf{k}'_{n,j} e^{in\theta_j}\|}. \tag{18}$$

It is easy to derive that

$$\|\mathbf{k}'_{m,j} e^{im\theta_j}\| = \|\mathbf{k}'_{m,j}\|, \tag{19}$$
$$\|\mathbf{k}'_{n,j} e^{in\theta_j}\| = \|\mathbf{k}'_{n,j}\|, \tag{20}$$

since the exponential terms do not change the vectors' magnitude.

Then, substitute the complex forms of $\mathbf{k}'_{m,j}$ and $\mathbf{k}'_{n,j}$, $\mathbf{k}'_{m,j} = k'_{m,2j} + ik'_{m,2j+1}$ and $\mathbf{k}'_{n,j} = k'_{n,2j} + ik'_{n,2j+1}$, respectively, back into $\langle \mathbf{k}'_{m,j} e^{im\theta_j}, \mathbf{k}'_{n,j} e^{in\theta_j} \rangle$ and obtain

$$\langle \mathbf{k}'_{m,j} e^{im\theta_j}, \mathbf{k}'_{n,j} e^{in\theta_j} \rangle = \langle \left( k'_{m,2j} + ik'_{m,2j+1} \right) e^{im\theta_j}, \left( k'_{n,2j} + ik'_{n,2j+1} \right) e^{in\theta_j} \rangle. \tag{21}$$

From Euler equation, Equation 21 can be further expanded as

$$
\begin{aligned}
\langle \mathbf{k}'_{m,j} & e^{im\theta_j}, \mathbf{k}'_{n,j} e^{in\theta_j} \rangle \\
= & k'_{m,2j} k'_{n,2j} \cos(m\theta_j) \cos(n\theta_j) + k'_{m,2j+1} k'_{n,2j+1} \sin(m\theta_j) \sin(n\theta_j) \\
& - k'_{m,2j} k'_{n,2j+1} \cos(m\theta_j) \sin(n\theta_j) - k'_{m,2j+1} k'_{n,2j} \sin(m\theta_j) \cos(n\theta_j) \\
& + k'_{m,2j} k'_{n,2j} \sin(m\theta_j) \sin(n\theta_j) + k'_{m,2j+1} k'_{n,2j+1} \cos(m\theta_j) \cos(n\theta_j) \\
& + k'_{m,2j} k'_{n,2j+1} \sin(m\theta_j) \cos(n\theta_j) + k'_{m,2j+1} k'_{n,2j} \cos(m\theta_j) \sin(n\theta_j) \\
= & k'_{m,2j} k'_{n,2j} \cos[(m-n)\theta_j] + k'_{m,2j+1} k'_{n,2j+1} \cos[(m-n)\theta_j] \\
& + k'_{m,2j} k'_{n,2j+1} \sin[(m-n)\theta_j] + k'_{m,2j+1} k'_{n,2j} \sin[(m-n)\theta_j].
\end{aligned}
\tag{22}
$$

Substitute Equation 22 back into Equation 18, and

$$
\begin{aligned}
\frac{\langle \mathbf{k}_{m,j}, \mathbf{k}_{n,j} \rangle}{\|\mathbf{k}_{m,j}\| \cdot \|\mathbf{k}_{n,j}\|} = & \frac{k'_{m,2j} k'_{n,2j} + k'_{m,2j+1} k'_{n,2j+1}}{\|\mathbf{k}'_{m,j}\| \cdot \|\mathbf{k}'_{n,j}\|} \cos[(m-n)\theta_j] \\
& + \frac{k'_{m,2j} k'_{n,2j+1} - k'_{m,2j+1} k'_{n,2j}}{\|\mathbf{k}'_{m,j}\| \cdot \|\mathbf{k}'_{n,j}\|} \sin[(m-n)\theta_j] \\
= & \frac{\mathbf{k}'_{m,j} \cdot \mathbf{k}'_{n,j}}{\|\mathbf{k}'_{m,j}\| \cdot \|\mathbf{k}'_{n,j}\|} \cos[(m-n)\theta_j] + \frac{\mathbf{k}'_{m,j} \times \mathbf{k}'_{n,j}}{\|\mathbf{k}'_{m,j}\| \cdot \|\mathbf{k}'_{n,j}\|} \sin[(m-n)\theta_j].
\end{aligned}
\tag{23}
$$

Let $\phi$ be angle between $\mathbf{k}'_{m,j}$ and $\mathbf{k}'_{n,j}$, then Equation 23 can be rewrite as

$$
\begin{aligned}
\frac{\langle \mathbf{k}_{m,j}, \mathbf{k}_{n,j} \rangle}{\|\mathbf{k}_{m,j}\| \cdot \|\mathbf{k}_{n,j}\|} = & \cos\phi \cos[(m-n)\theta_j] + \sin\phi \sin[(m-n)\theta_j] \\
= & \cos[\phi - (m-n)\theta_j].
\end{aligned}
\tag{24}
$$

Since the similarity between $\mathbf{k}_{m,j}$ and $\mathbf{k}_{n,j}$ nearly equals 1 as we assumed, it can be obtained that
$$
\phi = (m-n)\theta_j.
\tag{25}
$$
From Equation 17,
$$
0 < \phi \le m - n, \quad \text{if } m > n,
\tag{26}
$$
$$
m - n \le \phi < 0, \qquad \text{if } m < n.
\tag{27}
$$
As a result,

$$
\cos(m-n) < \frac{\langle \mathbf{k}'_{m,j}, \mathbf{k}'_{n,j} \rangle}{\|\mathbf{k}'_{m,j}\| \cdot \|\mathbf{k}'_{n,j}\|} \le 1.
\tag{28}
$$

## A.2 HIGH COSINE SIMILARITY AMONG KEY STATES

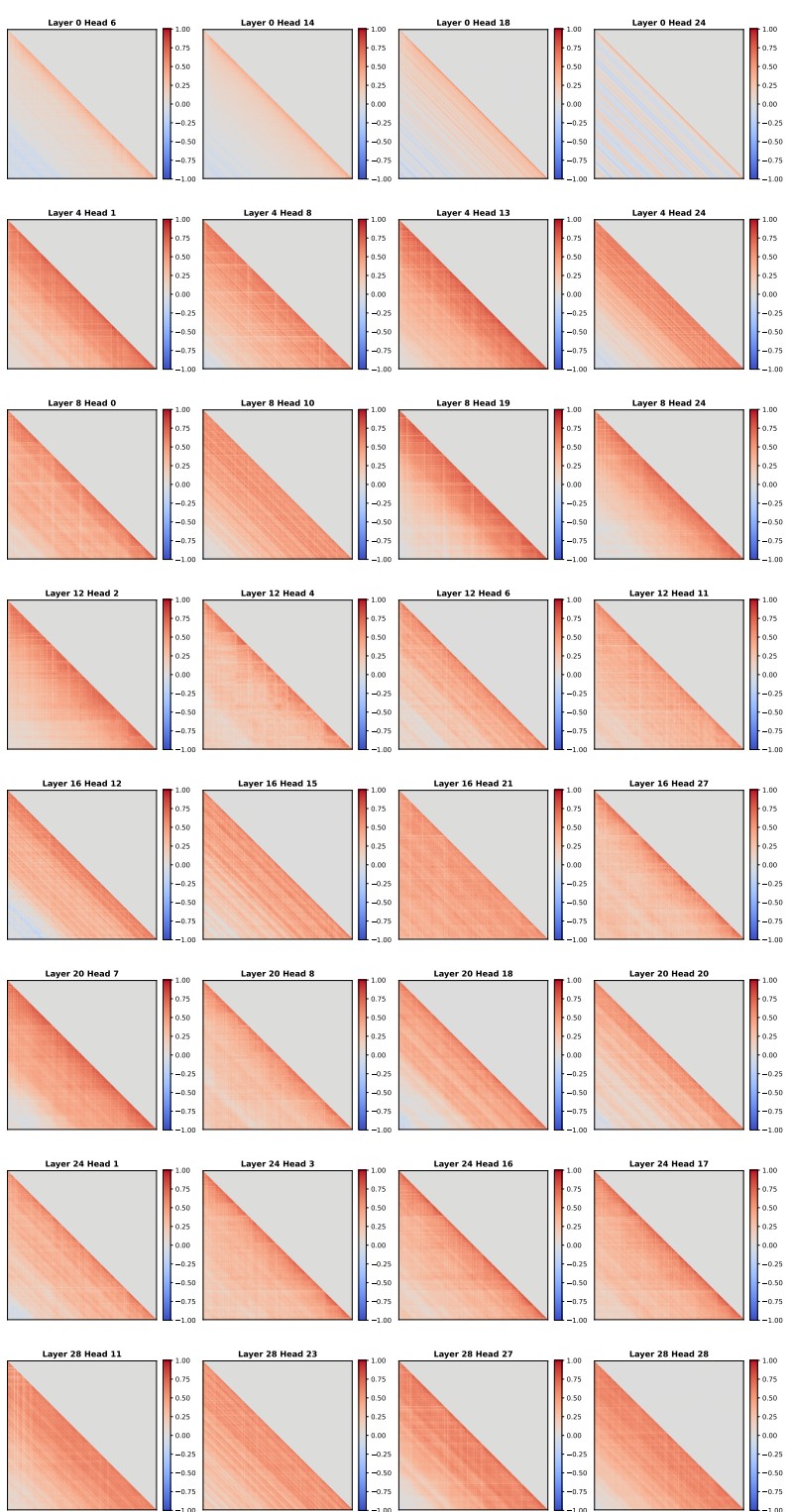

Figure 6: Visualization on the cosine similarity map of key states across different layers and attention heads in **Llama2-7B-chat** model when processing the input sequence with a total of 3506 tokens.

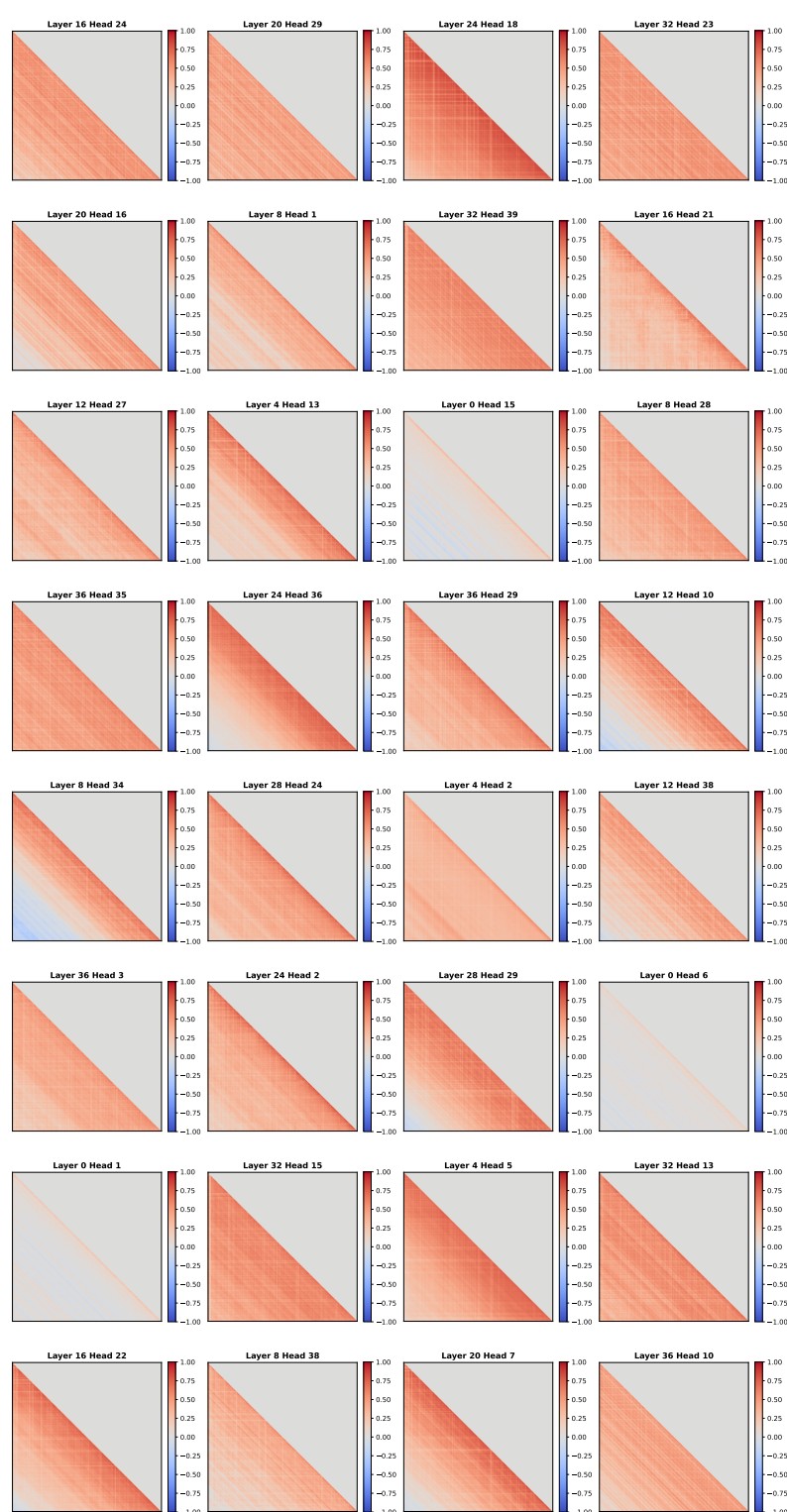

Figure 7: Visualization on the cosine similarity map of key states across different layers and attention heads in **Llama2-13B-chat** model when processing the input sequence with a total of 3776 tokens.

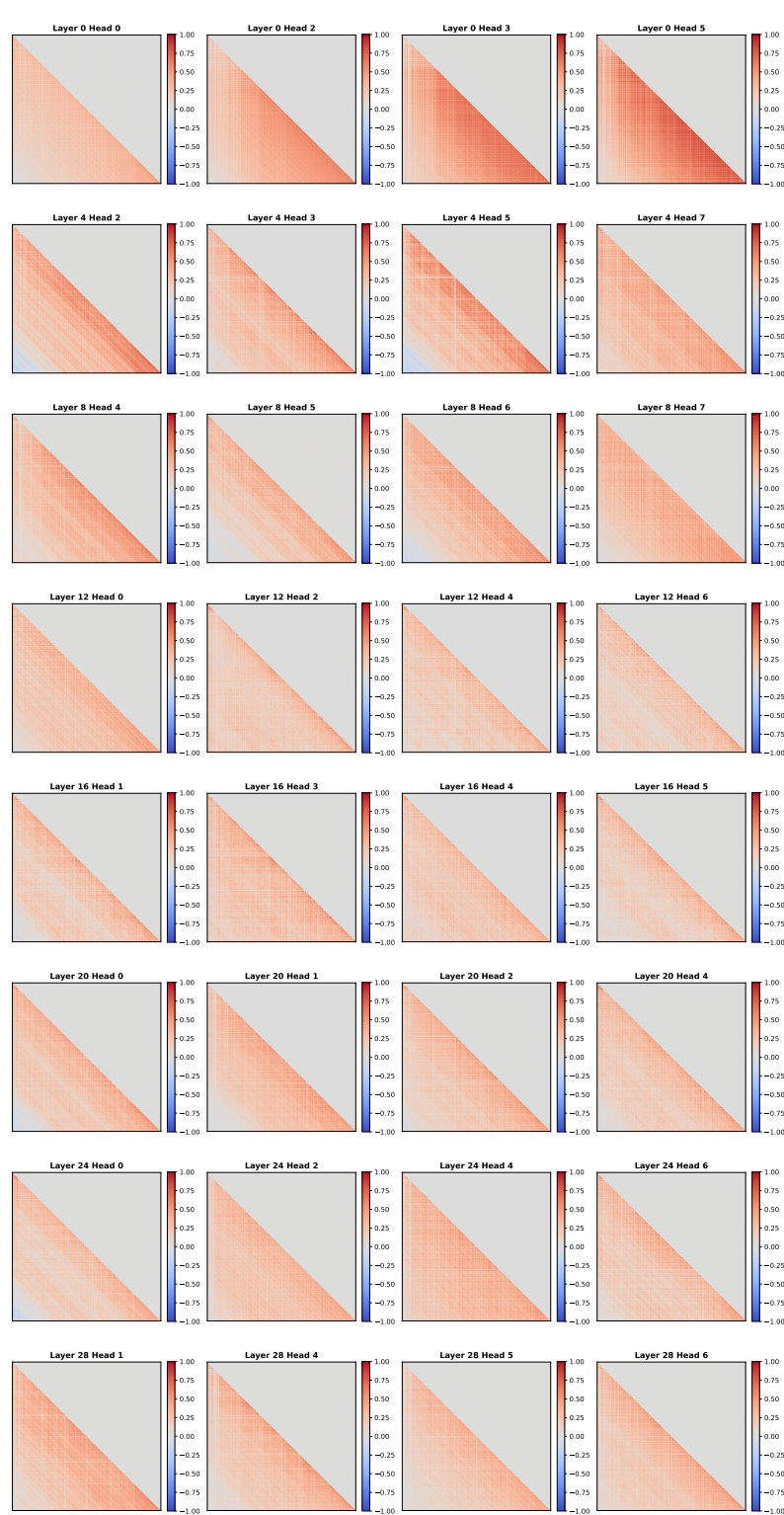

Figure 8: Visualization on the cosine similarity map of key states across different layers and attention heads in **Mistral-7B-Instruct** model when processing the input sequence with a total of 5001 tokens.

## A.3  STATIC KV CACHE SPARSITY

In section 4.2, we observe that the KV cache sparsity, resulting from the high similarity exhibited by key states, is independent of the dataset and remains persistent at the model level. To further illustrate this property, we also visualize the layer-wise compression ratios of Llama2-13B-chat and Mistral-7B-Instruct obtained by our proposed merging set identification algorithm for some selected tasks from LongBench, which is shown in Figure 9. We can observe that the static KV cache sparsity property also holds true for these models, making it convenient for us to set the memory budget for inference without significant adjustment.

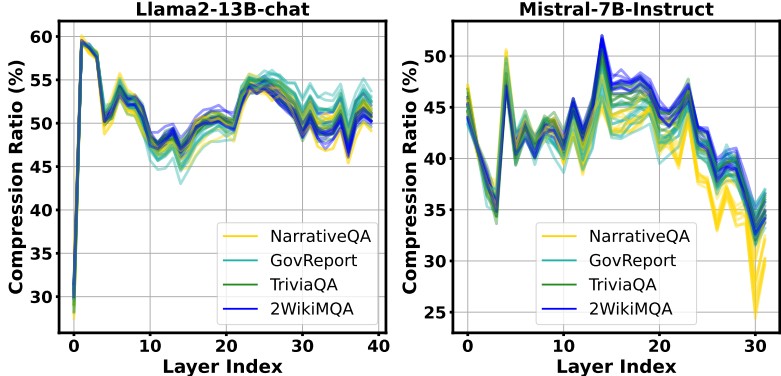

Figure 9: The layer-wise compression ratios of Llama2-13B-chat and Mistral-7B-Instruct obtained by our proposed merging set identification algorithm for different samples and different tasks.

## A.4  ZEROSCROLLS BENCHMARK RESULTS

We provide the evaluation results of Llama2-7B-chat model on ZeroScrolls Benchmark with *KVMerger*. From Table 6, we can spot that *KVMerger* achieves excellent performance compared to H2O and CaM methods under the same KV cache budget scenarios, with some results even better than the condition under full cache. This emphasize that *KVMerger* can effectively maintain the long-context processing ability of LLMs with compressed KV cache.

Table 6: *KVMerger* for Llama2-7B-chat on selected **ZeroScrolls** datasets

| cache budget | Method | gov_report | SummScreenFD | QMSum | SQuALITY | Qasper | NarrativeQA | BookSumSort | avg. |
|---|---|---|---|---|---|---|---|---|---|
| 100% | Full Cache | 17.40 | 14.10 | 15.20 | 19.50 | 22.50 | 15.40 | 3.00 | 15.30 |
| 50% | H2O | 15.40 | 13.20 | 14.30 | 18.30 | 20.50 | 15.00 | **3.80** | 14.36 |
| | CaM | 15.60 | 13.10 | 13.70 | 18.50 | 20.10 | 15.30 | 3.40 | 14.24 |
| | *KVMerger* | **17.70** | **13.80** | **15.10** | **19.10** | **22.50** | **15.20** | 3.10 | **15.21** |
| 35% | H2O | 14.80 | 11.60 | 14.20 | 17.80 | 17.70 | 14.70 | 3.60 | 13.49 |
| | CaM | 15.30 | 11.70 | 13.90 | 18.30 | 17.10 | 14.50 | 3.30 | 13.44 |
| | *KVMerger* | **16.60** | **13.80** | **15.40** | **18.60** | **20.40** | **15.40** | **3.70** | **14.84** |

