# OpenReview forum: "Model Tells You Where to Merge: Adaptive KV Cache Merging for LLMs on Long-Context Tasks"
_ICLR.cc/2025/Conference — Submitted to ICLR 2025_

### Official Review · Reviewer_7eNt · 2024-10-28

**Soundness:** 3
**Presentation:** 3
**Contribution:** 3
**Rating:** 3
**Confidence:** 4

**Summary:**

Goal is to reduce the size of the KV-cache. The main approaches for this include quantization, cache eviction, and state merging. The core observation is that key states have high cosine similarity at the token level within a single sequence across different heads and model layers. The proposed approach cluster states, then uses Gaussian kernel weighted merging function to merge the elements within set. The method also retains anything with a top-k attention score (“sinks” style). The method is evaluated on a series of LongBench tasks, compared to CaM and H2O prior methods.

**Strengths:**

- The method is simple to implement, well-motivated, and is explained clearly
- The paper compares to popular methods like H2O and CaM and offers strong improvements over them -- for this reason, the paper has the potential to be a significant step forwards
- The ablations around the choice of $\sigma$ and choice of merging algorithm are convincing

**Weaknesses:**

The claim that RoPE is required to achieve high similarity is not well supported. It is not clear why RoPE is required to meet the stated conditions in Lemmas 4.1 and 4.2.
- The paper shows that key states exhibit similarity, but suggests that value states do not (Lines 203-205); it would be good to support this claim with experiments

The method’s analysis could be improved
- It would be useful to further characterize when the method is expected to work well, as a function of the model and dataset – the analysis is incomplete. The current claim that the compression ratio remains the same independent of the dataset and model level seems incautious (Lines 267-283) – for instance, Mistral vs. Llama show very different trends in Table 1.
- Most of the evaluated benchmarks are memorization oriented: rather than testing how well the model uses the context, models could use information stored in the MLP weights to succeed on most of these benchmarks (e.g., Wiki, QA, news tasks). It would be useful to consider a broader set of in-context learning tasks.
- Supporting this point that the chosen tasks are memorization-oriented, all the KV-cache compression methods work well on Llama models, and Llama is trained on many more tokens vs. Mistral.

The method is not compared to quantization and eviction methods; the paper suggests that all these categories methods tackle the same overarching objective, so it is important to compare to those methods’ quality as well.

The method does not explain how much of the gain is due to including the tokens with the top-k attention scores (Line 323). Do H2O and CaM do this as well? If not, then that could explain the improvement over those methods. I could not find this point clearly explained in the paper.

**Questions:**

There are several missing experiments and analyses -- see weaknesses.

---

> ### Author Response · Authors · 2024-11-25
> **Response to Reviewer 7eNt**
>
> We greatly appreciate your review efforts. Thank you for your encouraging recognition of KVMerger’s potential to be a significant step forward! Below, we address your questions/comments and provide detailed clarifications:
>
> ### **W1. The claim that RoPE is required to achieve high similarity is not well supported. It is not clear why RoPE is required to meet the stated conditions in Lemmas 4.1 and 4.2.**
>
> **R1:**
> Thank you for this thoughtful question. We would like to clarify the role of RoPE in enhancing cosine similarity between two key states and how this relates to these two lemmas.
>
> **1. Clarifying the Role of RoPE**
>
> - RoPE is not required to meet the conditions in Lemmas 4.1 and 4.2. Rather, these lemmas describe the conditions under which RoPE can help two key states achieve high cosine similarity after its application. The key takeaway is that RoPE facilitates the preservation and enhancement of cosine similarity between key states to 1 when they already possess certain properties described in Lemma 4.2.
>
>
> **2. Interpreting Lemma 4.1**
>
> - Lemma 4.1 states that if the cosine similarity between two key state vectors $k_n$ and $k_m$ is equal to 1, then the cosine similarity between each corresponding sub-vector pair within  $k_n$ and $k_m$ is also 1. This implies that perfect similarity at the overall vector level leads to perfect similarity at the sub-vector level. The sub-vector level similarity is the condition stated in Lemma 4.2, which is particularly important because the RoPE operation operates on sub-vector pairs for each key state.
>
>
>
> **3. Interpreting Lemma 4.2**
>
> - Lemma 4.2 indicates that if the cosine similarity between the corresponding sub-vector pairs at the same position in $k_n$ and $k_m$ is 1 after the RoPE operation, then the cosine similarity of these sub-vector pairs before RoPE must fall within a specific bound, rather than being a random value. This indicates that RoPE creates a relationship between the perfect post-RoPE similarity (cosine similarity equals 1) and a bounded, non-random pre-RoPE similarity. Furthermore, even if the pre-RoPE similarity of a sub-vector pair falls outside the established bounds, RoPE can still enhance its post-RoPE similarity, bringing it closer to 1.
>
> **4. Evidence from Empirical Observation**
>
> - The necessity of RoPE in achieving high cosine similarity between two key states is also supported by empirical observation, which is shown by the figures in the link: https://imgur.com/a/aGPkn4x.
>
> - This result illustrates the similarity map of key states at a specific layer of the Llama2-7B-chat model while processing an input sequence of approximately 2000 tokens. Notably, before the RoPE operation (left figure), the key states exhibit low cosine similarity among one another. However, after applying RoPE (right figure), the key states demonstrate a distinct local high similarity pattern, highlighting the transformative effect of RoPE in enhancing cosine similarity for key states.
>
> ### **W2. The paper shows that key states exhibit similarity, but suggests that value states do not (Lines 203-205); it would be good to support this claim with experiments**
>
> **R2:**
>
> Thank you for raising this excellent point. Below, we provide a detailed explanation and additional support for our claim:
>
> **1. Key and Query States vs. Value States**
>
> - According to the RoPE paper [1], RoPE is applied exclusively to key and query states. As a result, both key and query states are expected to exhibit high similarity after the application of RoPE, as demonstrated in our analysis. In contrast, RoPE is not applied to value states and therefore value states do not undergo the same transformation, nor do they exhibit this property.
>
> **2. Visualization of Cosine Similarity Maps**
> - To further support this distinction, we visualized the cosine similarity maps for both query states (figure link:  https://imgur.com/a/ydVinpy) and value states (figure link: https://imgur.com/a/VRrsDYn) at certain layers and heads of the Llama2-7B-chat model while processing an input sequence of approximately 2000 tokens.
>
> - The results confirm that query states, like key states, exhibit a high degree of similarity following the RoPE operation. In contrast, value states, which do not apply RoPE, do not demonstrate this behavior. These findings provide compelling empirical evidence supporting our theoretical analysis of RoPE's cosine similarity enhancement properties.

---

> ### Author Response · Authors · 2024-11-25
>
> ### **W3. It would be useful to further characterize when the method is expected to work well, as a function of the model and dataset – the analysis is incomplete. The current claim that the compression ratio remains the same independent of the dataset and model level seems incautious (Lines 267-283) – for instance, Mistral vs. Llama show very different trends in Table 1.**
>
> **R3:**
>
> Thank you for bringing up this important point, and we apologize for any confusion may have caused.
>
> - We would like to clarify that when we state that the KV cache sparsity for different samples are persistent at the model level, we mean that **for each individual model, the layer-wise compression ratios achieved by our merging set identification process remain consistent across various datasets**, as shown in Figure 3b and Appendix A3 in the manuscript.
> - We acknowledge that different models exhibit different layer-wise compression ratios, which explains the varying trends observed between Mistral and Llama in Table 1.
>
> ### **W4. Most of the evaluated benchmarks are memorization oriented: rather than testing how well the model uses the context, models could use information stored in the MLP weights to succeed on most of these benchmarks (e.g., Wiki, QA, news tasks). It would be useful to consider a broader set of in-context learning tasks.**
>
> **R4:**
>
> - Thank you for raising this insightful point regarding the nature of the evaluated benchmarks and the potential reliance on memorization through MLP weights. We acknowledge that memorization can contribute to model performance on certain tasks. However, we believe that the benchmarks employed in our study go beyond mere memorization for several reasons:
>
> **1. Diversity and Complexity of LongBench**
>
> - The LongBench used in our experiments has been universally used in evaluating LLMs’ long context processing ability. It encompasses a wide range of tasks, including single or multi document QA, summarization, few-shot learning, retrieval task, and etc. These tasks require complex reasoning, understanding, and in-context processing, which goes beyond simple fact retrieval from MLP weights.
>
> **2. Significance of KVMerger on LongBench**
>
> - Enhancing LLM performance on LongBench tasks using a compressed KV cache is inherently challenging because it requires maintaining contextual integrity and supporting complex reasoning that cannot rely solely on MLP weights. Our experiments demonstrate that KVMerger consistently outperforms H2O and CaM on LongBench tasks, highlighting that KVMerger not only preserves but also enhances the model’s ability to effectively utilize context, rather than depending on memorized data.
>
> **3. In-context learning tasks**
>
> - LongBench includes in-context learning tasks such as **TREC** and **TriviaQA**, which require deep reasoning and effective context utilization. Our results in Table 1 clearly show that KVMerger outperforms H2O and CaM on these tasks. This also demonstrates that improving LLM performance on long-context tasks is not solely achievable through memorization via MLP weights but relies on advanced contextual processing, validating the effectiveness of KVMerger.
>
> ### **W5. Supporting this point that the chosen tasks are memorization-oriented, all the KV-cache compression methods work well on Llama models, and Llama is trained on many more tokens vs. Mistral.**
>
> **R5:**
>
> - As we discussed in R4, LongBench incorporates a wide range of tasks that assess not only memorization but also the model’s ability to utilize context effectively. Specifically, LongBench includes in-context learning tasks such as TREC and TriviaQA, which require deep reasoning and contextual understanding rather than mere recall of memorized information.
>
> - Although Llama is trained on a larger corpus of tokens than Mistral, KVMerger consistently outperforms other compression methods (H2O and CaM) across both models. This demonstrates that **KVMerger’s advantages are not solely due to the volume of training data**. Instead, KVMerger effectively optimizes the KV cache to enhance contextual processing across different model architectures and training regimes, highlighting its robustness and adaptability in improving context utilization regardless of the underlying model’s training scale.

---

> ### Author Response · Authors · 2024-11-25
>
> ### **W6. The method is not compared to quantization and eviction methods; the paper suggests that all these categories methods tackle the same overarching objective, so it is important to compare to those methods’ quality as well.**
>
> **R6:**
>
> We appreciate the reviewer’s suggestion to compare KVMerger with quantization and eviction methods. While both categories aim to optimize KV cache utilization, KVMerger specifically **addresses the limitations inherent in eviction-based approaches**.
>
> **1. Focus on Eviction-Based Comparisons**
>
> - Eviction-based methods often result in the loss of critical contextual information necessary for complex reasoning tasks. KVMerger is designed to preserve essential contextual data more effectively than traditional eviction strategies.
>
> - This focus is why our primary comparisons have been with **eviction-based method like H2O**, demonstrating that KVMerger maintains and enhances contextual integrity while optimizing cache size.
>
> **2. Quantization Methods Overview**
> - Although KV cache quantization introduces information loss by reducing the precision of key-value representations, it generally maintains a balance between efficiency and fidelity.
>
> - However, quantization primarily targets memory footprint reduction and does not directly address the preservation of contextual relationships within the KV cache.
>
> **3. More Comprehensive Comparison**
>
> - To provide a more comprehensive understanding of KVMerger’s relative performance and illustrate its effectiveness compared with other KV cache compression method, we have expanded our comparisons to include three additional eviction-based KV cache compression techniques: PyramidInfer[2], PyramidKV[3], and SnapKV[4].  The corresponding results are detailed below:

---

> ### Author Response · Authors · 2024-11-25
>
> | **Models**         | **KV Cache %** | **Method**        | **2wikimqa** | **gov_report** | **narrativeqa** | **pr_en** | **multifieldqa_en** | **trec** | **multi_news** | **triviaqa** | **qasper** | **avg**        |
> |---------------------|----------------|-------------------|--------------|----------------|-----------------|-----------|---------------------|----------|---------------|--------------|------------|----------------|
> | Llama2-7B-chat      | 100%           | Full Cache        | 31.45        | 26.99          | 18.74           | 8.00         | 36.60               | 64.00       | 26.20          | 83.09        | 21.83      | 35.22    |
> |                     |  50%               | KVMerger          | 32.99    | 25.31          | 18.50       | 7.33      | 36.89          | 64.18| 26.20          | 83.62    | 20.04      | **35.02**    |
> |                     |  50%               | PyramidInfer      | 31.53        | 25.39      | 17.87           | 8.50       | 35.36              | 64.00       | 26.20          | 83.26        | 21.02      | 34.80   |
> |                     |  50%               | PyramidKV         | 30.68        | 25.51          | 17.70            | 11.00        | 35.35              | 64.00       | 26.90      | 82.76        | 20.54      | 34.94    |
> |                     | 50%            | SnapKV            | 31.00           | 25.03          | 17.95           | 11.50      | 36.03              | 64.00       | 26.50          | 82.93        | 20.18      | 35.02    |
> |                     | 35%            | KVMerger          | 32.29        | 25.24          | 19.12           | 7.00         | 33.82              | 63.50     | 25.60          | 82.76        | 21.09      | **34.50**   |
> |                     |   35%              | PyramidInfer      | 29.81        | 25.23          | 18.62           | 7.00         | 35.00                 | 63.50     | 26.20          | 82.26        | 19.08      | 34.08    |
> |                     |    35%             | PyramidKV         | 29.36        | 25.44          | 16.72           | 11.00        | 34.73              | 64.00       | 25.50          | 82.42        | 18.73      | 34.22    |
> |                     |   35%              | SnapKV            | 28.87        | 23.97          | 15.53           | 11.00        | 35.45              | 64.00       | 25.40          | 82.03        | 19.95      | 34.03          |
> | Llama2-13B-chat     | 100%           | Full Cache        | 13.21        | 27.59          | 14.42           | 15.25     | 27.44              | 68.50     | 26.60          | 87.42        | 17.15      | 33.07    |
> |                     |   50%             | KVMerger          | 13.46        | 26.63          | 14.40            | 16.00        | 27.29              | 68.50     | 26.10          | 87.48        | 17.22      | **33.01**   |
> |                     |   50%             | PyramidInfer      | 13.63        | 26.29          | 14.47           | 15.50      | 27.22              | 68.00       | 25.50          | 86.73        | 17.09      | 32.72    |
> |                     |    50%            | PyramidKV         | 13.54        | 25.43          | 14.81           | 15.50      | 27.64              | 68.50     | 25.50          | 87.88        | 17.03      | 32.88          |
> |                     | 50%            | SnapKV            | 13.57        | 25.04          | 14.69           | 15.00        | 27.35              | 68.50     | 25.50          | 87.54        | 17.16      | 32.72    |
> |                     | 35%            | KVMerger          | 12.61        | 26.12          | 13.60            | 14.00        | 26.75              | 68.00       | 26.30          | 86.76        | 16.24      | **32.27**    |
> |                     |     35%            | PyramidInfer      | 13.19        | 26.29          | 13.49           | 14.55     | 26.45              | 67.50     | 25.50          | 87.03        | 16.21      | 32.25    |
> |                     |  35%               | PyramidKV         | 13.18        | 25.96          | 13.34           | 14.00        | 26.31              | 68.50     | 25.50          | 87.58        | 16.17      | 32.26    |
> |                     |   35%              | SnapKV            | 13.11        | 25.16          | 13.85           | 14.00        | 26.23              | 68.50     | 25.70          | 86.30         | 16.48      | 32.15    |
>
> - According to the above table, KVMerger demonstrates comparable or better performance across various tasks in LongBench compared to other eviction-based KV cache compression methods. These results underscore KVMerger's effectiveness as a promising solution for KV cache compression in LLMs without sacrificing overall accuracy. We will add this results in the final version of our paper.

---

> ### Author Response · Authors · 2024-11-25
>
> ### **W7. The method does not explain how much of the gain is due to including the tokens with the top-k attention scores (Line 323). Do H2O and CaM do this as well? If not, then that could explain the improvement over those methods. I could not find this point clearly explained in the paper.**
>
> **R7:**
>
> Thank you for this perceptive inquiry! We appreciate the opportunity to clarify this aspect of our work.
>
> **1. Role of Top-K Attention Scores in KVMerger**
>
> - Our KVMerger method strategically excludes tokens with the top-k attention scores from merging to prioritize the most contextually relevant information during KV cache compression.
> - This approach ensures that essential tokens, which significantly contribute to the model's reasoning and contextual understanding, are preserved.
>
> **2. Comparison with H2O and CaM**
>
> - H2O leverages top-k attention scores for KV cache compression by **selecting the top-k KV pairs based on their aggregated attention scores** to retain the most relevant information.
>
> - CaM focuses solely on merging value states while leaving the key states unmerged. Consequently, **CaM retains all key states** to preserve the information.
>
> - In comparison, KVMerger not only retains the top-k KV pairs based on attention scores but also **employs an advanced merging strategy** that effectively combines similar KV pairs. This dual approach optimizes cache utilization while maintaining essential contextual information more effectively than both H2O and CaM with reduced KV cache memory.
>
> **3. Empirical Evidence Supporting KVMerger’s Superiority**
>
> - Our experimental results, presented in Table 1 and additional results in response R6, demonstrate that KVMerger consistently outperforms both H2O and CaM, as well as other eviction-based methods which also inherently retain the top-k KV pairs, on long-context learning tasks.
>
> - This superior performance indicates that the gains achieved by KVMerger are not solely attributable to the inclusion of top‑k attention scores but also to its effective merging mechanism.

---

> ### Author Response · Authors · 2024-11-25
>
> **Reference**
>
> [1] Su, J., Lu, Y., Pan, S., Murtadha, A., Wen, B., & Liu, Y. (2023). RoFormer: Enhanced Transformer with Rotary Position Embedding. arXiv preprint arXiv:2104.09864.
>
> [2] Yang, D., Han, X., Gao, Y., Hu, Y., Zhang, S., & Zhao, H. (2024). PyramidInfer: Pyramid KV Cache Compression for High-throughput LLM Inference. arXiv preprint arXiv:2405.12532.
>
> [3] Zhang, Y., Gao, B., Liu, T., Lu, K., Xiong, W., Dong, Y., ... & Xiao, W. (2024). PyramidKV: Dynamic KV Cache Compression based on Pyramidal Information Funneling. arXiv preprint arXiv:2406.02069.
>
> [4] Li, Y., Huang, Y., Yang, B., Venkitesh, B., Locatelli, A., Ye, H., ... & Chen, D. (2024). Snapkv: Llm knows what you are looking for before generation. arXiv preprint arXiv:2404.14469.

---

> ### Author Response · Authors · 2024-12-01
>
> Dear Reviewer 7eNt,
>
> Thank you once again for the time and effort you've invested in reviewing our paper. We would like to kindly remind you that we have diligently addressed each point raised in your review.
>
> As the deadline for the author-reviewer discussion period is fast approaching, we are eagerly awaiting your feedback. We would be more than happy to address any additional concerns or comments you may have.
>
> Thank you!
>
> Best Regards,
> Authors of Paper #12608

---

### Official Review · Reviewer_F84t · 2024-11-03

**Soundness:** 3
**Presentation:** 2
**Contribution:** 3
**Rating:** 5
**Confidence:** 4

**Summary:**

This paper introduces KVMerger, a KV cache merging algorithm that adaptively compresses KV cache without significant performance loss. KVMerger uses a merging set identification algorithm and Gaussian kernel weighted merging to selectively merge similar KV states. Experiments on various LLMs show KVMerger outperforms baselines.

**Strengths:**

1. The writing is easy to follow.
2. The proposed method seems concise and effective.

**Weaknesses:**

1. The proof in Sec 4.1 makes the story confusing. In Sec 4.1, it is proven that the value states have low similarity due to the lack of RoPE representation, so why does your method still merge the value states and it is also effective?
2. Some experimental details are missing. Your Merging Set Identification likely requires calculating the similarity of states on a dataset. Which dataset do you use?
3. Some baselines are missing, such as PyramidInfer[1], PyramidKV[2] and SnapKV[3].
4. Is your merging only performed during the prefill stage? During generation, is merging also applied?
5. It would be best to compare the throughput during inference.
6. The equal sign '=' in Eq. (5) is used incorrectly. This is a formula, and '=' is not an assignment operator. You could represent the merged key state as something like $k_p^{*}$.

[1] Yang, D., Han, X., Gao, Y., Hu, Y., Zhang, S., & Zhao, H. (2024). PyramidInfer: Pyramid KV Cache Compression for High-throughput LLM Inference. arXiv preprint arXiv:2405.12532.

[2] Zhang, Y., Gao, B., Liu, T., Lu, K., Xiong, W., Dong, Y., ... & Xiao, W. (2024). PyramidKV: Dynamic KV Cache Compression based on Pyramidal Information Funneling. arXiv preprint arXiv:2406.02069.

[3] Li, Y., Huang, Y., Yang, B., Venkitesh, B., Locatelli, A., Ye, H., ... & Chen, D. (2024). Snapkv: Llm knows what you are looking for before generation. arXiv preprint arXiv:2404.14469.

**Questions:**

1. Your objective function in Lines 311-314 appears to be NP-hard. How close is the solution found by your Algorithm 1 to this objective function?

---

> ### Author Response · Authors · 2024-11-25
> **Response to Reviewer F84t**
>
> We greatly appreciate your review efforts to our work. Thank you for your encouraging recognition of KVMerger’s effectiveness in long-context tasks with compressed KV cache for LLMs! Below, we address your questions/comments and provide detailed clarifications:
>
> ### **W1. In Sec 4.1, it is proven that the value states have low similarity due to the lack of RoPE representation, so why does your method still merge the value states, and it is also effective?**
>
> **R1:**
>
> - Thank you for this insightful question. During the initial development of KVMerger, we deliberately avoided merging value states and still achieved results comparable to the full KV cache scenario.
>
> - However, we empirically found that merging value states does not hurt accuracy but significantly improves the compression ratio. This echoes findings from D2O [1] and CaM [2]. We hypothesize that value states are inherently less sensitive to merging due to redundancy in their representations. Even with relatively low similarity, value states often encode overlapping or related token information, allowing them to be compressed without significantly affecting task performance.
>
> - Our evaluation results on LongBench tasks across various models reflect the effectiveness of this adjustment. We plan to include results of both with and without value state merging in the final version to provide a comprehensive view of its impact.
>
> ### **W2. Your Merging Set Identification likely requires calculating the similarity of states on a dataset. Which dataset do you use?**
>
> **R2:**
>
> - Thank you for raising this question. We sincerely apologize for any confusion regarding the merging set identification process. To clarify, our method does not rely on a pre-defined calibration dataset to calculate the similarity of states.
>
> - Instead, the merging set identification process is conducted **adaptively on-the-fly**. Specifically, for each input sample, the similarity calculation is performed directly during runtime based on the current key states generated by the model. This ensures that the merging decisions are tailored to the specific characteristics of each input, maximizing the flexibility and effectiveness of our approach.
>
> ### **W3. Some baselines are missing, such as PyramidInfer, PyramidKV and SnapKV.**
>
> **R3:**
>
> Thank you for this valuable suggestion. To provide a more comprehensive understanding of KVMerger’s relative performance and illustrate its effectiveness compared with other KV cache compression method, we have expanded our comparisons to include three additional KV cache compression techniques: PyramidInfer, PyramidKV, and SnapKV. The corresponding results are detailed below:

---

> ### Author Response · Authors · 2024-11-25
>
> | **Models**         | **KV Cache %** | **Method**        | **2wikimqa** | **gov_report** | **narrativeqa** | **pr_en** | **multifieldqa_en** | **trec** | **multi_news** | **triviaqa** | **qasper** | **avg**        |
> |---------------------|----------------|-------------------|--------------|----------------|-----------------|-----------|---------------------|----------|---------------|--------------|------------|----------------|
> | Llama2-7B-chat      | 100%           | Full Cache        | 31.45        | 26.99          | 18.74           | 8.00         | 36.60               | 64.00       | 26.20          | 83.09        | 21.83      | 35.22    |
> |                     |  50%               | KVMerger          | 32.99    | 25.31          | 18.50       | 7.33      | 36.89          | 64.18| 26.20          | 83.62    | 20.04      | **35.02**    |
> |                     |  50%               | PyramidInfer      | 31.53        | 25.39      | 17.87           | 8.50       | 35.36              | 64.00       | 26.20          | 83.26        | 21.02      | 34.80   |
> |                     |  50%               | PyramidKV         | 30.68        | 25.51          | 17.70            | 11.00        | 35.35              | 64.00       | 26.90      | 82.76        | 20.54      | 34.94    |
> |                     | 50%            | SnapKV            | 31.00           | 25.03          | 17.95           | 11.50      | 36.03              | 64.00       | 26.50          | 82.93        | 20.18      | 35.02    |
> |                     | 35%            | KVMerger          | 32.29        | 25.24          | 19.12           | 7.00         | 33.82              | 63.50     | 25.60          | 82.76        | 21.09      | **34.50**   |
> |                     |   35%              | PyramidInfer      | 29.81        | 25.23          | 18.62           | 7.00         | 35.00                 | 63.50     | 26.20          | 82.26        | 19.08      | 34.08    |
> |                     |    35%             | PyramidKV         | 29.36        | 25.44          | 16.72           | 11.00        | 34.73              | 64.00       | 25.50          | 82.42        | 18.73      | 34.22    |
> |                     |   35%              | SnapKV            | 28.87        | 23.97          | 15.53           | 11.00        | 35.45              | 64.00       | 25.40          | 82.03        | 19.95      | 34.03          |
> | Llama2-13B-chat     | 100%           | Full Cache        | 13.21        | 27.59          | 14.42           | 15.25     | 27.44              | 68.50     | 26.60          | 87.42        | 17.15      | 33.07    |
> |                     |   50%             | KVMerger          | 13.46        | 26.63          | 14.40            | 16.00        | 27.29              | 68.50     | 26.10          | 87.48        | 17.22      | **33.01**   |
> |                     |   50%             | PyramidInfer      | 13.63        | 26.29          | 14.47           | 15.50      | 27.22              | 68.00       | 25.50          | 86.73        | 17.09      | 32.72    |
> |                     |    50%            | PyramidKV         | 13.54        | 25.43          | 14.81           | 15.50      | 27.64              | 68.50     | 25.50          | 87.88        | 17.03      | 32.88          |
> |                     | 50%            | SnapKV            | 13.57        | 25.04          | 14.69           | 15.00        | 27.35              | 68.50     | 25.50          | 87.54        | 17.16      | 32.72    |
> |                     | 35%            | KVMerger          | 12.61        | 26.12          | 13.60            | 14.00        | 26.75              | 68.00       | 26.30          | 86.76        | 16.24      | **32.27**    |
> |                     |     35%            | PyramidInfer      | 13.19        | 26.29          | 13.49           | 14.55     | 26.45              | 67.50     | 25.50          | 87.03        | 16.21      | 32.25    |
> |                     |  35%               | PyramidKV         | 13.18        | 25.96          | 13.34           | 14.00        | 26.31              | 68.50     | 25.50          | 87.58        | 16.17      | 32.26    |
> |                     |   35%              | SnapKV            | 13.11        | 25.16          | 13.85           | 14.00        | 26.23              | 68.50     | 25.70          | 86.30         | 16.48      | 32.15    |
>
> - According to the above table, KVMerger demonstrates comparable or better performance across various tasks in LongBench compared to other KV cache compression methods. These results underscore KVMerger's effectiveness as a promising solution for KV cache compression in large language models without sacrificing overall accuracy. We will add this results in the final version of our paper.

---

> ### Author Response · Authors · 2024-11-25
>
> ### **W4. Is your merging only performed during the prefill stage? During generation, is merging also applied?**
>
> **R4:**
>
> - Thank you for this insightful question! KVMerger is specifically designed for long-context reasoning tasks, where the primary bottleneck typically lies in managing the extensive context length during the prefill stage.
>
> - The total number of tokens processed during the generation stage is usually quite small relative to the input context in such tasks. For example, in LongBench tasks, sample lengths are consistently greater than 4,000 tokens, with some even reaching up to 10,000 tokens.
>
> - In contrast, the generated outputs average no more than 512 tokens, and sometimes are as short as less than 10 tokens. Therefore, KVMerger are mainly applied at the prefilling stage, addressing the most pressing performance and memory challenges while maintaining the efficiency and effectiveness of the model.
>
> ### **W5. It would be best to compare the throughput during inference.**
>
> **R5:**
>
> Thank you for highlighting this important aspect of our analysis! The current implementation of KVMerger does not yield throughput improvements. To provide clarity, we measured the latency on the Llama2-7B-chat model using a context length of 4000 and a generation length of 1024 on an A100 GPU. The results are presented as follows:
>
> |                         | **Full Cache (s)**      | **KVMerger with 50% Cache (s)**     |
> |-------------------------|---------------------|------------------|
> | **Prefilling Stage**    | 0.463               | 14.651            |
> | **Decoding Stage**| 26.635             | 20.637           |
> | **Total Latency**       | 27.135             | 35.338           |
>
> - For each layer of LLM, we also measured the latencies for different components of KVMerger during the prefilling stage. The breakdown is as follows:
>
> |                                | **Cosine Similarity Calculation** | **Merging Set Identification** | **Gaussian Kernel Merging** |
> |--------------------------------|-----------------------------------|---------------------------------|-----------------------------|
> | **Latency (ms)**               | 1.507                            | 443.641                         | 0.148                      |
>
>
> - We can see that for the current implementation, merging set identification is a significant bottleneck for KVMerger compared to full-cache scenarios, as shown in the table above. This process, executed on the CPU, involves two nested for-loops and has an asymptotic computational complexity of $O(nh)$, where $n$ is the context length and $h$ is the number of attention heads.
>
> - However, this overhead can be addressed by optimizing implementation. A promising solution is to parallelize the merging process across layers so that the latency for merging set identification can be minimized, leading to the improved overall throughput.
>
> ### **W6. The equal sign '=' in Eq. (5) is used incorrectly. This is a formula, and '=' is not an assignment operator. You could represent the merged key state as something like $k_{p}^{*}$.**
>
> **R6:**
>
> Thank you for pointing out this. We have modified it in our manuscript.

---

> ### Author Response · Authors · 2024-11-25
>
> ### **Q1: Your objective function in Lines 311-314 appears to be NP-hard. How close is the solution found by your Algorithm 1 to this objective function?**
>
> **R:**
>
> - Thank you very much for this insightful question! Directly comparing the solution produced by Algorithm 1 to the optimal value of the NP-hard objective function defined in Definition 5.1 is infeasible for large instances, as the time required to solve such NP-hard problems grows exponentially with the number of tokens. However, we can approximate the quality of the solution using a heuristic based on intra-cluster and inter-cluster similarities:
>   - **Intra-cluster similarity**: measures how well similar key states are grouped together for all merging set, which corresponds to the first item in the objective function in Definition 5.1;
>   - **Inter-cluster similarity**: measures how well dissimilar key states are separated into distinct merging sets, which corresponds to the second item in the objective function in Definition 5.1.
>
> - Specifically, we apply Algorithm 1 with three different cosine similarity thresholds to the Llama2-7B-chat model, using an input sequence length of 4000. Algorithm 1 determines the merging sets for each head at each layer, as illustrated in Figure 4 of the manuscript. We then compute the intra- and inter-cluster similarities for each head and each layer in terms of the above definitions. Then, we take the average values of these computed intra- and inter-cluster similarities, and the results are presented in the following table:
>
> | **Threshold** | **Intra Similarity** | **Inter Similarity**  |
> |---------------|-----------|------------|
> | **0.75**      | 0.8339    | 0.0828     |
> | **0.85**      | 0.8919    | 0.0830   |
> | **0.95**      | 0.9578    | 0.0830   |
>
> - The table demonstrates that the intra-cluster similarity is significantly higher than the inter-cluster similarity across various similarity thresholds. Furthermore, the intra-cluster similarity increases as the cosine threshold is raised, indicating that the similarity threshold directly influences the quality of intra-cluster cohesion.
>
> - As the thresholds increase, the intra-cluster similarity improves, while the inter-cluster similarity remains relatively constant. This suggests that the separation between clusters is not significantly affected by the threshold, and the primary driver of the difference between intra- and inter-cluster similarity is the increase in intra-cluster similarity.
>
> - Maximizing the difference between intra- and inter-cluster similarity is equivalent to maximizing the intra-cluster similarity under the given conditions. This observation highlights that the merging sets generated by Algorithm 1 exhibit the desired behavior of maximizing the objective function outlined in Definition 5.1, effectively balancing intra-cluster similarity and inter-cluster separation.

---

> ### Author Response · Authors · 2024-11-25
>
> **Reference**
>
> [1] Wan, Z., Wu, X., Zhang, Y., Xin, Y., Tao, C., Zhu, Z., Wang, X., Luo, S., Xiong, J., & Zhang, M. (2024). D2O: Dynamic Discriminative Operations for Efficient Generative Inference of Large Language Models. arXiv. Retrieved from https://arxiv.org/abs/2406.13035
>
> [2] Zhang, Y., Du, Y., Luo, G., Zhong, Y., Zhang, Z., Liu, S., & Ji, R. (2024). CAM: Cache merging for memory-efficient LLMs inference. In Proceedings of the 41st International Conference on Machine Learning (ICML).

---

> > ### Comment · Reviewer_F84t · 2024-11-26
> >
> > Thanks for your clarification and supplementary experiments. The current manuscript is not ready for publication, so I will keep my score.

---

> > > ### Author Response · Authors · 2024-11-27
> > >
> > > Thank you very much for your prompt feedback and for acknowledging our clarifications and additional experiments. We would greatly appreciate **any specific suggestions on how we can further improve the manuscript to make it ready for publication**. Thank you again for your time and valuable feedback.

---

### Official Review · Reviewer_AQkX · 2024-11-04

**Soundness:** 3
**Presentation:** 3
**Contribution:** 2
**Rating:** 5
**Confidence:** 4

**Summary:**

The paper proposes KVMerger, an adaptive KV cache merging algorithm for large language models (LLMs) performing long-context tasks. By identifying and merging highly similar key-value (KV) cache states, the approach aims to reduce memory usage without compromising model performance. The authors use cosine similarity to cluster similar KV states and employ a Gaussian kernel-weighted merging technique. The effectiveness of KVMerger is demonstrated on various benchmarks and LLMs, with promising results compared to existing KV cache compression methods like H2O and CaM.

**Strengths:**

+ The paper introduces a unique perspective by formulating KV cache merging as a constrained clustering problem, presenting a fresh method to tackle memory efficiency challenges in long-context LLM tasks
+ KVMerger empirically appears to achieve notable memory savings, as evidenced by reduced GPU memory consumption during inference

**Weaknesses:**

- The paper lacks a thorough discussion on the limitations or potential drawbacks of KVMerger, such as scenarios where high similarity assumptions in KV cache may fail or lead to degraded performance
- The merging algorithm’s reliance on cosine similarity computations and Gaussian kernel-weighted merging could be computationally expensive, especially for very large models, yet the paper provides limited insight into the computational overhead involved
- The paper assumes that highly similar key states contain redundant information, but it lacks a robust analysis of the potential impact of merging on nuanced context understanding, which is critical for certain tasks
- The study only compares KVMerger with H2O and CaM. Including more baselines or a hybrid method combining quantization with merging would better situate KVMerger’s relative effectiveness

**Questions:**

See weakness

---

> ### Author Response · Authors · 2024-11-25
> **Response to Reviewer AQkX**
>
> We greatly appreciate your review efforts. Thank you for your encouraging recognition of KVMerger’s promising results in maintaining the performance of LLMs on long-context tasks with a compressed KV cache. Below, we address the potential weaknesses you pointed out and provide detailed clarifications:
>
> ### **W1: A thorough discussion on the limitations or potential drawbacks of KVMerger is needed.**
>
> **R1:**
>
> Thank you very much for this suggestion! We fully recognize the importance of discussing the limitations of KVMerger. Below, we provide additional points discussing the potential limitations of KVMerger:
>
> 1. **Generalization Ability of KVMerger**
>
>    - Most state-of-the-art LLMs, including the **Llama2 series**, **Llama3 series**, and **Mistral series**, adopt **Rotary Position Embedding (RoPE)** for position encoding. This design inherently causes the key states to exhibit high similarity, as demonstrated through our theoretical and empirical analyses in the paper. Consequently, KVMerger can be directly applied to most LLMs.
>
>    - To further validate its effectiveness, we also applied KVMerger to the **Llama3-8B-Instruct-8k** model. The results, shown below, demonstrate that KVMerger outperforms methods such as **H2O** and **CaM**, achieving the best results across various long-context tasks. These results underscore its robust generalization capabilities, which will be included in the final version of the paper.
>
> | **KV Cache Budget** | **Method**   | **2wikimqa** | **gov_report** | **narrativeqa** | **pr_en** | **multifieldqa_en** | **trec** | **multi_news** | **triviaqa** | **qasper** | **avg**          |
> |------------|--------------|--------------|----------------|-----------------|-----------|---------------------|----------|---------------|--------------|------------|------------------|
> | 100%       | Full Cache   | 34.77    | 28.67      | 25.53       | 69.25 | 39.22          | 73.5 | 26.64     | 90.48    | 32.03  | 46.68 |
> | 50%        | KVMerger     | **34.87**    | **26.86**      | **25.71**       | **69.25** | 37.38          | **72.00**   | **25.66**     | **90.46**    | 27.81  | **45.56** |
> | 50%           | H2O          | 33.88        | 26.58          | 22.69           | 66.83     | 37.44              | 72.00      | 24.90          | 90.41        | **28.96**      | 44.85     |
> | 50%           | CaM          | 32.95        | 26.37          | 24.19           | 65.25     | **37.73**          | 72.00       | 25.40          | 90.29        | 28.27      | 44.72     |
> | 35%        | KVMerger     | **34.58**    | **27.46**      | **24.45**       | **68.5**  | **37.10**           | **71.50** | **26.04**     | **90.13**    | **28.18**  | **45.32** |
> | 35%            | H2O          | 33.86        | 26.73          | 23.55           | 67.00        | 36.07              | 70.50     | 25.91         | 90.10         | 28.02      | 44.64     |
> |  35%           | CaM          | 34.15        | 26.53          | 23.30            | 67.67     | 36.92              | 71.50     | 25.92         | 89.51        | 27.85      | 44.82     |
>
> 2. **Acknowledging Limitations of KVMerger**
>
> Despite its numerous advantages, KVMerger has certain limitations that need further exploration and refinement. These include:
>
> - **Incompatibility with Flash Attention**
>   - KVMerger cannot integrate with Flash Attention, as it relies on computing attention scores dynamically during the merging process. Flash Attention, which optimizes memory efficiency by computing attention scores in parallel without explicitly materializing the attention matrix, is inherently incompatible with the current merging mechanism of KVMerger.
>   - To address this, future research could explore alternative merging techniques or adaptations without relying on attention scores to make KVMerger compatible with memory-efficient attention mechanisms while preserving its performance benefits.
>
> - **Performance Gap**
>   - Current KV cache compression methods, including KVMerger, cannot fully recover the original performance of LLMs across all tasks. This limitation highlights an intrinsic trade-off between memory efficiency and task performance in KV cache compression. However, we want to emphasize that **KVMerger still achieves the most compelling performance compared to other methods in terms of our results**.
>
> - **Latency Overhead**
>   - The merging set identification process introduce additional latency compared to a full-cache scenario during the prefilling stages. This overhead could impact real-time or latency-sensitive applications.
>   - To mitigate this issue, exploring more efficient merging set identification algorithms would be a promising direction. Additionally, precomputing certain merging operations during the initialization phase could help reduce runtime latency.
>
> Despite these limitations, we believe KVMerger represents a significant step forward in enabling efficient KV cache compression for long-context tasks in LLMs.

---

> ### Author Response · Authors · 2024-11-25
>
> ### **W2: Computational Overhead Analysis for KVMerger**
>
> **R2:**
> - Thank you for highlighting the potential overhead introduced by KVMerger. To better understand the latency impact of the KVMerger process, we measured the latency on the Llama2-7B-chat model using an A100 GPU. The experiment was conducted with a context length of 4000 and a generation length of 1024. The latency results are summarized below:
>
> |                         | **Full Cache (s)**      | **KVMerger with 50% Cache (s)**     |
> |-------------------------|---------------------|------------------|
> | **Prefilling Stage**    | 0.463               | 14.651            |
> | **Decoding Stage**| 26.635             | 20.637           |
> | **Total Latency**       | 27.135             | 35.338           |
>
> - For each layer of LLM, we also measured the latencies for different components of KVMerger during the prefilling stage. The breakdown is as follows:
>
> |                                | **Cosine Similarity Calculation** | **Merging Set Identification** | **Gaussian Kernel Merging** |
> |--------------------------------|-----------------------------------|---------------------------------|-----------------------------|
> | **Latency (ms)**               | 1.507                            | 443.641                         | 0.148                      |
>
>
> - We can see that for the current implementation, merging set identification is a significant bottleneck for KVMerger compared to full-cache scenarios, as shown in the table above. This process, executed on the CPU, involves two nested for-loops and has an asymptotic computational complexity of $O(nh)$, where $n$ is the context length and $h$ is the number of attention heads.
>
> - However, this overhead can be addressed by optimizing implementation. A promising solution is to parallelize the merging process across layers. This approach could effectively eliminate the bottleneck and substantially reduce latency, enabling KVMerger to scale more efficiently.
>
> ### **W3: Robust analysis of the potential impact of merging on nuanced context understanding**
> **R3:**
>
> Thank you for this insightful question! We recognize that nuanced context understanding is crucial for certain tasks, and we appreciate the opportunity to clarify how our method addresses this concern.
>
> - We retain the top-k KV states with the highest aggregated attention scores, including both attention sinks and heavy-hitters important for overall comprehension. Previous eviction-based methods like H2O discard KV states with lower attention scores, which often contain nuanced information. This leads to significant performance drops in tasks requiring subtle understanding, such as **multifieldqa_en** and **gov_report** in the LongBench benchmark (please refer to Table 1 in our paper).
>
> - In contrast, KVMerger mitigates this issue by merging these lower-scoring KV states instead of discarding them. This merging process preserves the essential subtle information embedded in the nuanced tokens.
>
> - While further analysis could provide deeper insights into the impact of merging on nuanced context understanding, our empirical results in Table 1 in the paper demonstrate that KVMerger effectively maintains or even enhances performance in tasks where nuanced understanding is critical.

---

> ### Author Response · Authors · 2024-11-25
>
> ### **W4: More baselines or a hybrid method combining quantization with merging would better situate KVMerger’s relative effectiveness**
> **R4:**
>
> Thank you for this valuable suggestion! To provide a more comprehensive understanding of KVMerger’s relative performance and illustrate its effectiveness compared with other KV cache compression methods, we have expanded our comparisons to include three additional KV cache compression techniques: PyramidInfer [1], PyramidKV [2], and SnapKV [3]. The corresponding results are detailed below:
> | **Models**         | **KV Cache %** | **Method**        | **2wikimqa** | **gov_report** | **narrativeqa** | **pr_en** | **multifieldqa_en** | **trec** | **multi_news** | **triviaqa** | **qasper** | **avg**        |
> |---------------------|----------------|-------------------|--------------|----------------|-----------------|-----------|---------------------|----------|---------------|--------------|------------|----------------|
> | Llama2-7B-chat      | 100%           | Full Cache        | 31.45        | 26.99          | 18.74           | 8.00         | 36.60               | 64.00       | 26.20          | 83.09        | 21.83      | 35.22    |
> |                     |  50%               | KVMerger          | 32.99    | 25.31          | 18.50       | 7.33      | 36.89          | 64.18| 26.20          | 83.62    | 20.04      | **35.02**    |
> |                     |  50%               | PyramidInfer      | 31.53        | 25.39      | 17.87           | 8.50       | 35.36              | 64.00       | 26.20          | 83.26        | 21.02      | 34.80   |
> |                     |  50%               | PyramidKV         | 30.68        | 25.51          | 17.70            | 11.00        | 35.35              | 64.00       | 26.90      | 82.76        | 20.54      | 34.94    |
> |                     | 50%            | SnapKV            | 31.00           | 25.03          | 17.95           | 11.50      | 36.03              | 64.00       | 26.50          | 82.93        | 20.18      | 35.02    |
> |                     | 35%            | KVMerger          | 32.29        | 25.24          | 19.12           | 7.00         | 33.82              | 63.50     | 25.60          | 82.76        | 21.09      | **34.50**   |
> |                     |   35%              | PyramidInfer      | 29.81        | 25.23          | 18.62           | 7.00         | 35.00                 | 63.50     | 26.20          | 82.26        | 19.08      | 34.08    |
> |                     |    35%             | PyramidKV         | 29.36        | 25.44          | 16.72           | 11.00        | 34.73              | 64.00       | 25.50          | 82.42        | 18.73      | 34.22    |
> |                     |   35%              | SnapKV            | 28.87        | 23.97          | 15.53           | 11.00        | 35.45              | 64.00       | 25.40          | 82.03        | 19.95      | 34.03          |
> | Llama2-13B-chat     | 100%           | Full Cache        | 13.21        | 27.59          | 14.42           | 15.25     | 27.44              | 68.50     | 26.60          | 87.42        | 17.15      | 33.07    |
> |                     |   50%             | KVMerger          | 13.46        | 26.63          | 14.40            | 16.00        | 27.29              | 68.50     | 26.10          | 87.48        | 17.22      | **33.01**   |
> |                     |   50%             | PyramidInfer      | 13.63        | 26.29          | 14.47           | 15.50      | 27.22              | 68.00       | 25.50          | 86.73        | 17.09      | 32.72    |
> |                     |    50%            | PyramidKV         | 13.54        | 25.43          | 14.81           | 15.50      | 27.64              | 68.50     | 25.50          | 87.88        | 17.03      | 32.88          |
> |                     | 50%            | SnapKV            | 13.57        | 25.04          | 14.69           | 15.00        | 27.35              | 68.50     | 25.50          | 87.54        | 17.16      | 32.72    |
> |                     | 35%            | KVMerger          | 12.61        | 26.12          | 13.60            | 14.00        | 26.75              | 68.00       | 26.30          | 86.76        | 16.24      | **32.27**    |
> |                     |     35%            | PyramidInfer      | 13.19        | 26.29          | 13.49           | 14.55     | 26.45              | 67.50     | 25.50          | 87.03        | 16.21      | 32.25    |
> |                     |  35%               | PyramidKV         | 13.18        | 25.96          | 13.34           | 14.00        | 26.31              | 68.50     | 25.50          | 87.58        | 16.17      | 32.26    |
> |                     |   35%              | SnapKV            | 13.11        | 25.16          | 13.85           | 14.00        | 26.23              | 68.50     | 25.70          | 86.30         | 16.48      | 32.15    |

---

> ### Author Response · Authors · 2024-11-25
>
> - According to the above table, KVMerger demonstrates comparable or better performance across various tasks in LongBench compared to other KV cache compression methods. These results underscore KVMerger's effectiveness as a promising solution for KV cache compression in large language models without sacrificing overall accuracy. We will add this results in the final version of our paper.
>
> **Reference**
>
> [1] Yang, D., Han, X., Gao, Y., Hu, Y., Zhang, S., & Zhao, H. (2024). PyramidInfer: Pyramid KV Cache Compression for High-throughput LLM Inference. arXiv preprint arXiv:2405.12532.
>
> [2] Zhang, Y., Gao, B., Liu, T., Lu, K., Xiong, W., Dong, Y., ... & Xiao, W. (2024). PyramidKV: Dynamic KV Cache Compression based on Pyramidal Information Funneling. arXiv preprint arXiv:2406.02069.
>
> [3] Li, Y., Huang, Y., Yang, B., Venkitesh, B., Locatelli, A., Ye, H., ... & Chen, D. (2024). Snapkv: Llm knows what you are looking for before generation. arXiv preprint arXiv:2404.14469.

---

> ### Author Response · Authors · 2024-12-01
>
> Dear Reviewer AQkX,
>
> Thank you once again for the time and effort you've invested in reviewing our paper. We would like to kindly remind you that we have diligently addressed each point raised in your review.
>
> As the deadline for the author-reviewer discussion period is fast approaching, we are eagerly awaiting your feedback. We would be more than happy to address any additional concerns or comments you may have.
>
> Thank you!
>
> Best Regards,
> Authors of Paper #12608

---

### Meta-Review · Area_Chair_njJ7 · 2024-12-20

**Metareview:**

## Summary:
The work proposes KVMerger to compress the KV cache and reduce memory consumption. Motivated by the high similarity (redundancy) between key states observed across different datasets on LLMs using RoPE, KVMerger first identifies the merging sets by solving a constrained clustering problem using a greedy algorithm and then merges all tokens in each set into one by Gaussian kernel weighted merging. It also includes the top-k tokens with the highest accumulated attention scores as previous eviction methods. Experiments on several long-context datasets show that KVMerger can achieve better or comparable performance as existing KV cache methods under different memory budget constraints.

## Strengths:
1. Most reviewers agree that the proposed strategy is well-motivated and presented clearly.
1. Promising improvement over baselines such as H2O and CaM.

## Weakness:
1. Reviewers are concerned about how general the assumptions of high similarity between key states and relatively lower similarity between value states hold, across different tasks and models.
1. Lack of comparisons to quantization and additional eviction methods for KV cache compression.
1. KVMerger may increase the latency or reduce the throughput.
1. Lack of ablation studies on (i) the contribution of memorization and (ii) the top-k attention-based eviction.

## Decision:
The authors responded to most of the concerns raised by the reviewers with additional experimental results, including the comparisons to three eviction baselines, KVMerger results on a new model, latency for different stages of the proposed method. They also provided more clarifications about the assumptions and generalizability. However, after carefully checking all the rebuttals, I feel that the following concerns still hold:
1. Comparisons to three additional baselines do not show significant advantages as in the comparisons to H2O and CaM. Under the same budgets, the performance improvement is marginal.
1. Lack of comparison to quantization baselines.
1. The latency caused by merging set identification is a weakness of the proposed method, though the authors mentioned some possible directions to improve it.
1. The ablation studies mentioned above are not provided, which may weaken the contribution of the proposed strategy.
1. Lack of experiments on other diverse tasks and datasets.

Considering the final review ratings, I cannot recommend acceptance of this paper to ICLR. The authors are encouraged to address the remaining concerns of the reviewers, improve the motivation part and the experimental part of this paper, and submit it to the next conference.

**Additional Comments On Reviewer Discussion:**

One out of the three reviewers participated in the discussion, confirming to keep the rating after reading the rebuttal. The meta-reviewer carefully read the author's responses and checked whether the points raised in the original reviews have been addressed or not. The final decision is mainly based on this feedback.

---

### Decision · Program_Chairs · 2025-01-22

Reject